# Revealing the sources and sinks of negative cluster ions in an urban environment through quantitative analysis

Rujing Yin[1,2#], Xiaoxiao Li[1,3#], Chao Yan[2,4], Runlong Cai[2], Ying Zhou[4], Juha Kangasluoma[2], Nina Sarnela[2], Janne Lampilahti[2], Tuukka Petäjä[2], Veli-Matti Kerminen[2], Federico Bianchi[2], Markku Kulmala[2,4], Jingkun Jiang[1*]

[1]State Key Joint Laboratory of Environment Simulation and Pollution Control, School of Environment, Tsinghua University, Beijing, 100084, China

[2]Institute for Atmospheric and Earth System Research/Physics, Faculty of Science, University of Helsinki, 00014 Helsinki, Finland

[3]School of Resources and Environmental Sciences, Wuhan University, 430072 Wuhan, China

[4]Aerosol and Haze Laboratory, Beijing Advanced Innovation Center for Soft Matter Science and Engineering, Beijing University of Chemical Technology, 100029 Beijing, China

[#]These authors contributed equally to this work.

[*]: *Correspondence to*: J. Jiang (jiangjk@tsinghua.edu.cn)

**This PDF file includes:**
  Main text
  Figures 1 to 10

**Abstract.** Atmospheric cluster ions are important constituents in the atmosphere, and their concentrations and compositions govern their role in atmospheric chemistry. However, there is currently limited quantitative research on atmospheric ion compositions, sources, and sinks, especially in the urban atmosphere where pollution levels and human populations are intense. In this study, we measured the compositions of negative cluster ions and neutral molecules using an atmospheric pressure interface high-resolution time-of-flight mass spectrometer (APi-TOF) and a chemical ionization mass spectrometer in urban Beijing. Quantitative analysis of cluster ions was performed by their comparison with condensation sink (CS), reagent ions, and neutral molecules. We demonstrate the feasibility of quantifying cluster ions with different compositions using *in-situ* measured ion mobility distributions from a neutral cluster and air ion spectrometer (NAIS). The median concentration of negative cluster ions was 85 (61-112 for 25-75%) $cm^{-3}$ during the measurement period, which was negatively correlated with CS. The negative cluster ions mainly consisted of inorganic nitrogen-containing ions, inorganic sulfur-containing ions, and organic ions in the form of adducts with $NO_3^-$ or $HSO_4^-$. The CHON-related organic ions accounted for over 70% of the total organic ions. Although the molecules clustered with $NO_3^-$ and $HSO_4^-$ had similar compositions, we found that $HSO_4^-$ clustered more efficiently with CHO and $CHON_{nonNPs}$ species (CHON excluding nitrated phenols), while $NO_3^-$ clustered more efficiently with nitrated phenols ($CHON_{nonNPs}$). Additionally, most organic ions were positively correlated with neutral molecules, resulting in similar diurnal cycles of organic ions and neutral molecules. However, an exception was found for $CHON_{NPs}$, the concentration of which is also significantly influenced by the reagent ions $NO_3^-$. The charge fractions are generally higher for molecules with higher molecular weight and oxidation state, and the opposite diurnal variations of charging fractions between $H_2SO_4$ and organic species indicate a charging competition between them. Finally, we choose $HSO_4^-$ and $C_3H_3O_4^-$ as representatives to calculate the contribution of different formation and loss pathways. We found their losses are condensational loss onto aerosol particles (73-75%), ion-molecule reaction losses (19%), and ion-ion recombination losses (6-8%).

**Keywords:** atmospheric cluster ions, ion composition, quantitative analysis, urban atmosphere, condensation sink

## 1 Introduction

Atmospheric cluster ions are electrically charged atoms, molecules, and molecular clusters. Their mobility is usually larger than 0.5 $cm^2 \cdot V^{-1} \cdot s^{-1}$ (Hõrrak et al., 2000). The primary ions, such as $N_2^+$, $O_2^+$, $NO^+$, $O^-$, and $O_2^-$, are initially formed via cosmic radiation, gamma radiation, and other near-ground local sources, such as radon decay, lightning, plant emissions, waterfall and seashore generation, combustion, and high-voltage transmission lines emission (Carslaw et al., 2002; Eisele, 1989; Hirsikko et al., 2007; Tammet et al., 2006). Primary ions will subsequently undergo ion-molecule reactions, ion-ion recombination, or deposition onto particles, accompanied by the evolution of their electrical mobility, concentration, and composition (Curtius et al., 2006; Harrison, 2003; Kontkanen et al., 2013; Shuman et al., 2015). Atmospheric ions play important roles in atmospheric chemistry (Bates, 1982; Luts and Salm, 1994), ion-induced nucleation (Charlson et al., 1992; Lee et al., 2003; Lovejoy, 2004), atmospheric conductivity (G. Baumgaertner et al., 2013) and air quality (Jiang et al., 2018). Moreover, atmospheric ions can benefit human health at certain levels of concentration, while their low concentrations may cause headaches and insomnia (Chu et al., 2019; Malcolm et al., 2009).

The effects of atmospheric cluster ions are highly related to their mobility, concentration, and composition.

There are substantial differences in the characteristics of atmospheric ions and their dynamic variations among diverse environments. Globally, atmospheric ion concentrations span a wide range from 100 to 5000 cm$^{-3}$ (Hirsikko et al., 2011; Usoskin et al., 2004), determined mainly by differences in ion production and loss rates. On one hand, ion production rates vary with geographical location, land cover type, and weather conditions (Chen et al., 2016; Ling et al., 2010; Usoskin et al., 2004). On the other hand, ion loss rates are largely related to ions carrying opposite charges, aerosol surface area concentrations, and deposition rates (Harrison, 2003; Tammet et al., 2006). For ion compositions, hundreds of new ions can be formed through ion-molecule reactions after the primary ions are initially formed. It has been found that besides the primary ions, $NO_3^-$, $HSO_4^-$, and their clusters were predominant ions due to their high acidities (Davidson et al., 1977; Viggiano et al., 1980). Oxygenated organic molecules (OOMs) were observed clustering with $NO_3^-$ and $HSO_4^-$ naturally in the atmosphere and the composition of these organic atmospheric ions has been reported to be similar to those of neutral vapor molecules (Bianchi et al., 2017; Ehn et al., 2010; Yin et al., 2021). Specifically, ion compositions were found very different between NPF and non-NPF events, and ion compositions during NPF imply different nucleation pathways in clean and polluted atmospheres (Bianchi et al., 2016; Ehn et al., 2010; Eisele et al., 2006; Kirkby et al., 2016; Yin et al., 2021). Thus, the measurement of ion characteristics requires high temporal measurements in various environments.

Ion characteristics at polluted urban sites, where the ion production and loss processes may be very different from the clean sites, are relatively poorly understood. At many polluted urban sites, intense pollution emissions led to high concentrations of neutral gas species and high aerosol concentrations. For the ion production processes, the neutral gas molecules at urban sites are significantly different from those at clean sites. High $NO_x$ concentrations were found to be accompanied by high concentrations of nitrogen-containing organic molecules at polluted urban sites (Li et al., 2022; Qiao et al., 2021). For ion loss processes, condensation sink (CS) at urban sites can reach up to two orders of magnitude higher values compared with clean sites (Cai et al., 2017), which would cause a significant loss of ions onto large particles in urban air. These may lead to big differences in ion concentrations and compositions between clean and urban sites. For instance, Iida et al. (2006) have shown that the ion concentration near a polluted site in Boulder might be substantially suppressed by high ion loss rates, resulting in only a minor contribution from ion-induced nucleation to NPF. Although some recent studies have revealed the characteristics of ions in clean environments (Bianchi et al., 2016; Chen et al., 2016; Ehn et al., 2010), only a few studies have focused on such characteristics in an urban atmosphere (Yin et al., 2021).

Simultaneous analysis of ion concentrations and compositions is rare in the field campaign. Various ion counters have been designed to measure ion concentrations in the last century. Among these, Gerdien counters (Gerdien, 1905), the balance scanning mobility analyzer (BSMA) (Tammet, 2006), the air ion spectrometer (AIS) (Mirme et al., 2007), and a neutral cluster and air ion spectrometer (NAIS) (Manninen et al., 2016; Manninen et al., 2009) are the most widely used ion counters or ion mobility spectrometers. However, these electrical mobility-based instruments do not provide composition information of ions. Mass spectrometers have been widely used to measure ion compositions. Quadrupole mass spectrometers with approximate unit mass resolution have been applied in a series of ion composition measurements (Arnold et al., 1978; Arnold and Viggiano, 1982; Eisele, 1989; Eisele et al., 2006; Eisele and Tanner, 1990;

Viggiano, 1993). With the development of high-resolution mass spectrometer techniques, such as an atmospheric pressure interface time-of-flight mass spectrometer (APi-TOF, Tofwerk AG), more species of atmospheric ions were identified (Ehn et al., 2010; Eisele et al., 2006; Junninen et al., 2010; Kirkby et al., 2016), but the concentration measured by the APi-TOF remains to be quantified.

It is needed to develop a robust and easy-to-use method to quantify concentrations of different ions in APi-TOF in field measurements. The detection efficiency of an APi-TOF to different ions is mainly determined by the mass-dependent transmission efficiency. The calibration of transmission efficiency often requires complex laboratory calibration settings, including the generation, classification, and counting of ions with different $m/z$ ratios (Heinritzi et al., 2016; Junninen et al., 2010). Alternatively, the relative transmission efficiency of an APi-TOF compared to primary ions can be obtained using a depletion method with the help of a chemical ionization inlet (Heinritzi et al., 2016). But the settings are still complex and only relative transmission efficiency is obtained. Ehn et al. (2011) have shown that the ion concentration measured by NAIS and the size-segregated ion signal measured by APi-TOF have good correlations in Hyytiälä. Then by converting the ion mobility into the $m/z$ of the ions (Hõrrak et al., 2000), the *in-situ* calibration of the mass spectrometers can be achieved using the synchronous measurements of well-calibrated ion spectrometers (e.g. NAIS). However, the feasibility of this method still needs to be examined, especially in polluted urban atmospheres.

In this study, we apply an *in-situ* quantification method of atmospheric ion compositions measured by an APi-TOF and reveal the governing factors of atmospheric cluster ion concentration and composition at polluted urban sites. We perform field measurements of atmospheric negative cluster ions at an urban site using the APi-TOF and NAIS simultaneously. By comparing the two *in-situ* measurements, we quantify the concentration of ions measured by the APi-TOF using an improved method. The concentration and composition of ions are compared with the clean sites, and the reasons for the observed differences are explored. The origins and composition variations of ions are characterized and compared with the neutral clusters measured by a chemical ionization mass spectrometer with $NO_3^-$ as the reagent ions (abbreviated as (nitrate) CI-APi-TOF). The driving factors for the variations in ion characteristics at the urban site are revealed, including CS, reagent ions, and neutral molecules. The charge fractions of various organic species were determined. To better quantify the formation and loss scheme of atmospheric ions, the ions $HSO_4^-$ and $C_3H_3O_4^-$ are taken as representatives in the mass balance equation and their variations are well-captured.

**2 Methods**

**2.1 Field measurement**

A field measurement is conducted at the Aerosol and Haze Laboratory of Beijing University of Chemical Technology Station (AHL/BUCT station), Beijing, China (Liu et al., 2020; Yin et al., 2021). The AHL/BUCT station is a typical urban site surrounded by three traffic roads and some residential buildings. The station is ~18 m above the ground with no higher buildings around it within 50 m. It is equipped with many state-of-the-art instruments and routine instruments for air quality measurements. Besides, the measurements at a boreal forest station SMEAR II (System for Measurement of forest Ecosystem and Atmospheric Relationships) located at Hyytiälä, Finland were used for comparison. Details of the instruments and their measurement periods in these two stations are given in Table S1.

The atmospheric negative cluster ion compositions were measured with an atmospheric pressure interface time-of-flight
high-resolution mass spectrometers (APi-HTOF) from Feb 14 to Feb 27, 2018, in Beijing, and from April 7 to June 8,
2013, in Hyytiälä (Yan et al., 2018). The ambient air was sampled from the window through a 1.4-m long 1/4 in, stainless
steel tubing. The sampling flow rate of the APi-HTOF is ~0.8 LPM, and an extra sheath flow of 3 LPM was sampled to
minimize the loss of ions in the sampling tubes. Thus the total sampling flow rate was 3.8 LPM. The APi-HTOF was
operated in the negative ion mode to measure negative ions and its voltages were adjusted to reduce cluster
fragmentations. The mass resolution of the APi-HTOF is ~4500 at $m/z$ 200. The elements C, H, O, N, and S were used
for peak assignment. Peaks that cannot be assigned with these elements within 10 ppm are labeled as "unknown" or
"others". The relative mass-dependent transmission efficiency in the HTOF was calibrated through the depletion method
when the APi-HTOF was operated with a nitrate chemical ionization (CI) inlet in front, for which the voltages were kept
the same as in the APi-mode (Heinritzi et al., 2016). Detailed descriptions of the measurement and data analysis have
been described in our previous study (Yin et al., 2021). In Hyytiälä, ions were sampled through a core sampling method.
It has a thick and long tube outside and a thin and short coaxial tube inside to minimize the sampling losses with a total
flow rate of 5 LPM. The details of the APi-TOF measurements in Hyytiälä can be found in a previous study (Yan et al.,
2018). For further analysis in this study, we quantified the negative cluster ions in both Beijing and Hyytiälä using the
*in-situ* quantification method, as will be discussed in Section 2.2.
The neutral gas molecules were measured by a (nitrate-) CI-APi-LTOF (Bertram et al., 2011; Jokinen et al., 2012) from
Jan 23 to April 14, 2018, in urban Beijing. It was operated side by side with the APi-TOF to achieve simultaneous
measurements of neutral molecules and cluster ions. $NO_3^-$ and its adducts $(HNO_3)_{1-2}NO_3^-$ were used as the reagent ions
to chemically ionize the gas molecules with a laminar flow ionization source mounted in front of the APi-LTOF (Eisele
and Tanner, 1993). Nitric acid is volatilized and carried by a total of 20 LPM sheath flow, then it is exposed to soft X-
ray to produce $NO_3^-$ and its adducts $(HNO_3)_{1-2}NO_3^-$. Mostly the species that can be ionized by $NO_3^-$ and its adducts
$(HNO_3)_{1-2}NO_3^-$ are highly oxygenated molecules, such as $H_2SO_4$ and OOMs. The mass resolution of the (nitrate-) CI-
APi-LTOF is ~8000 at $m/z$ 200. The peak fitting and assignment of the molecular compositions followed our previous
studies where binPMF solutions were used as assistants for the peak assignment (Nie et al., 2022). The sensitivity for
$H_2SO_4$ was calibrated via the controlled reaction between $SO_2$ and OH at the end of the campaign (Kürten et al., 2012;
Li et al., 2019). The relative mass-dependent transmission efficiency of the CI-APi-LTOF was calibrated using the
depletion method (Heinritzi et al., 2016). OOMs were quantified assuming they have the same collision-limit charging
efficiency as $H_2SO_4$ with reagent ions, and the mass-dependent transmission efficiency is also used for the quantification
of OOMs. The calibration results have been shown in our previous study (Yin et al., 2021).
The ion mobility distribution was measured by a NAIS (NAIS, Airel Ltd.) (Mirme and Mirme, 2013) from Jan 12[th] to
Dec 31[st], 2018 in urban Beijing, and from Jan 1[st] to Dec 31[st], 2013 in Hyytiälä (Kontkanen et al., 2013). In both stations,
the measurement periods of NAIS covered the measurement periods of APi-TOF. NAIS can simultaneously measure
positive and negative ions with ion mobility of 3.2-0.0013 $cm^2 \cdot V^{-1} \cdot s^{-1}$, corresponding to the mobility diameter range of
0.8-40 nm. In practice, the NAIS switched between detecting naturally charged ions and total particles (including neutral
and charged particles) by switching off/on a unipolar corona charger. The measurement cycle period was 2 min for the
ion mode, 2 min for the particle mode, and 30 s for offset. For the ion measurement, two cylindrical mobility

spectrometers were operated in parallel to measure positive and negative ions simultaneously. In this study, we focus on the negative cluster ions measured in the ion mode, whose ion mobility is larger than 0.8 cm$^2 \cdot$V$^{-1} \cdot$s$^{-1}$ and diameter smaller than ~1.6 nm. A total of 109 mobility bins in the negative ion mode were measured in urban Beijing, while a total of 28 mobility bins were measured in Hyytiälä. In urban Beijing, ambient air is drawn into the system through a 1.8 m long copper tube with a diameter of 4 cm, positioned at the window. The sample flow rate is maintained at 54 LPM. The inversion kernel of NAIS was calibrated based on the method described by Wagner et al. (2016) and sampling losses were further considered in the data inversion. The results of NAIS were calibrated considering both the transfer functions and sampling losses (Mirme and Mirme, 2013; Wagner et al., 2016).

**2.2 The quantification of negative cluster ion**

Although we have calibrated the relative mass-dependent transmission efficiency in the APi-HTOF, there are still two problems for quantification: (1) the absolute transmission efficiency is unknown; (2) the relative transmission efficiency curve is calibrated by adding a CI-inlet in front of the APi-HTOF, and the influences of CI-inlet remain unknown. For example, a previous study using a high-resolution differential mobility analyzer and electrometer to calibrate the transmission efficiencies under the APi-mode and CI-mode found that the transmission efficiency under the CI-mode shifts to larger $m/z$ compared to APi-mode (Heinritzi et al., 2016).

To address these two problems in quantifying APi-HTOF, we used an improved method to obtain the absolute transmission efficiency of the APi-HTOF through the comparison with the synchronous *in-situ* measurement from the NAIS. The key to linking the measurement results of ions by the APi-HTOF with those by the NAIS is to convert the $m/z$ of ions to ion mobilities. As the resolving power of APi-HTOF is higher than that of NAIS, the ion signals in APi-HTOF in specific $m/z$ ranges were summed up to compare with the ion concentrations measured by the NAIS.

For some specific ions with known peak assignments and structures, like $NO_3^-$ and $HSO_4^-$, the ion mobility can be directly measured by ion mobility spectrometers (IMS). The results reported in previous literature (Jen et al., 2015; Liang et al., 2013; Spangler and Collins, 1975; Stano et al., 2008) are summarized in Table S1 and were applied in this study. For the remaining ions, like the hundreds of complex organic ions, the Stokes-Millikan equation was applied to convert their $m/z$ to mobilities (Ehn et al., 2011; Friedlander, 2000; Tammet, 1995), as in Eq. 1.

$$Z = \frac{1}{\sqrt{1 + m_g / m}} \frac{Q}{3\pi\mu} \frac{1 + Kn\left(1.257 + 0.4e^{-1.1/Kn}\right)}{d_m + d_g}$$

Eq.1

where $Z$ is the electrical mobility, $m_g$ the mass of a carrier gas molecule, $m$ the mass of ions, $Q$ the charge of ions, $\mu$ the viscosity of the carrier gas, $Kn$ the Knudsen number, $d_m$ the mobility diameter of ions, and $d_g$ the diameter of the carrier gas molecule. Here, the carrier gas is ambient air, and $m_g$, $d_g$, and $\mu$ are adopted as 29.0 g/mol, 0.3 nm, and 1.73×10$^{-5}$ kg$\cdot$m$^{-1} \cdot$s$^{-1}$ at 278 K and 101.3 kPa (Ku and De La Mora, 2009). $Kn$ and $d_m$ are calculated as Eq. 2-3.

$$K_n = \frac{2\lambda}{d_m + d_g}$$

Eq.2

$$d_m = \sqrt[3]{\frac{6m}{\pi\rho}}$$

Eq.3

where $\lambda$ is the mean free path of carrier gas (62 nm at 278 K and 101.3 kPa) and $\rho$ the density of ions. The appropriate

density for calculating the mobility of ions is controversial. Ehn et al. (2011) used the bulk density of aconitic acid (1.66 g·cm$^{-3}$) for the conversion, but molecular clusters may have different densities with bulk substances. To address this problem, we compared the measured mobility of an IMS for different types of OOMs with the calculated mobilities according to the Stokes-Millikan method using 1.1, 1.3, and 1.6 g·cm$^{-3}$ as the densities, respectively (Fig. S1). The experimental results of the IMS are from Krechmer et al. (2016). By comparing the measured and calculated mobilities for organic ions with different compositions, densities of 1.1 and 1.3 g·cm$^{-3}$ were ultimately used in the mass-mobility conversion of ions with $m/z$ smaller than 150 or larger than 200, respectively, whereas between $m/z$ of 150 and 200, a linear curve of density is used. Though some ions with m/z larger than 400 fall into the region of 1.1 g·cm$^{-3}$ in Fig. S1, they were mostly OOM dimers that did not exist in Beijing (Qiao et al., 2021). Thus a density of 1.3 g·cm$^{-3}$ was applied for large organic ions in Beijing. Finally, the mobilities of all ions were adjusted according to the ambient temperature and pressure before being applied in the calibration of APi-HTOF (Tammet, 1995).

Using the synchronous *in-situ* measurement results of the NAIS as the reference, we obtained the detection efficiency of the APi-HTOF for ions with different $m/z$. Here, since the ion concentrations measured by the NAIS have already considered sampling losses and represent atmospheric concentrations, the obtained detection efficiency of the APi-HTOF should represent a combined result of the transmission efficiency of the APi-HTOF and the sampling efficiency in the sampling line of the APi-HTOF. The ions used in the APi-HTOF are in the $m/z$ range from 32 to 908, corresponding to the mobility diameters of 0.8-1.6 nm and ion mobilities of 3.16 to 0.77 cm$^2$·V$^{-1}$·s$^{-1}$, covering 12 mobility bins of the NAIS measurements. The number concentrations and signal intensities measured by the NAIS and APi-HTOF had good consistencies (r $\geq$ 0.66) in the 12 bins, and the correlations were best over the $m/z$ range of 331-791 (r > 0.94) (Fig. S2). The good correlation between the APi-HTOF and NAIS indicates that both instruments captured the variation of the atmospheric negative cluster ions. Then the detection efficiency of a certain $m/z$ in the APi-HTOF was calculated as the slope of the linear fitting line between signal intensities acquired in the APi-HTOF and number concentrations in the NAIS. Compared with the detection efficiencies calculated from the mass-dependent density, using a fixed density of 1.66 g·cm$^{-3}$ would overestimate the detection efficiency for small ions and underestimate that for large ions, with a deviation of up to 40% (Fig. S3).

After excluding the effect of sampling losses in the APi-HTOF, the absolute transmission efficiency of the APi-HTOF was derived, and the obtained pattern agreed well with the relative transmission efficiency as shown in Fig. 1. The sampling losses of ions in the APi-TOF is assumed the same as those of neutral particles with the same sizes (Mahfouz and Donahue, 2021). As the mass-dependent density was applied, the absolute and relative transmission efficiency curves showed similar patterns, i.e., both peaked at ~$m/z$ 400 and the maximum absolute transmission efficiency reached ~4.9%. The discrepancy between the absolute and relative transmission efficiency at small values of m/z may be caused by the uncertainty of small ion measurements in the NAIS, sampling efficiency calculation, or the influences of CI-inlet, as the voltage settings remain the same during the calibration experiment. It would result in an uncertainty of ~28% for the quantification of small ions. The absolute transmission efficiency obtained by our calibration is roughly the same order of magnitude as that in previous studies, and the shape is similar as well (Heinritzi et al., 2016; Junninen et al., 2010). Finally, the fit line of the detection efficiency was applied to quantify the ion compositions measured by the APi-HTOF, considering both the sampling efficiency and the transmission efficiency (Fig. 1).

Fig. 1 The detection and transmission efficiency curves

After the quantification, the total negative cluster ion concentrations measured by the APi-HTOF were consistent with those measured by the NAIS (Fig. 2). However, when the total ion concentration was lower than ~70 cm$^{-3}$, the total ion concentration measured by the NAIS was higher than that measured by the APi-HTOF. Specifically, the total cluster ion concentration measured by the NAIS was still higher than 25 cm$^{-3}$ even when ions detected by APi-HTOF approached zero. These signals detected by the NAIS are mainly from small cluster ions (3.16-1.28 cm$^2$·V$^{-1}$·s$^{-1}$, 0.8~1.27 nm), the fraction of which rises to >80% of the total counts when the total cluster ion concentration measured by the APi-HTOF is lower than ~70 cm$^{-3}$. As the APi-HTOF has a much lower detection limit than the NAIS (Fig. S4), these signals could be caused by the high noise of small ions in the NAIS. The detection limit of the APi-HTOF is the lowest at a diameter of 1.4 nm, which is ~0.02 cm$^{-3}$, and it increases when the detected ions move to lower or higher sizes due to the decrease in detection efficiency. Under a minimum diameter of 0.9 nm and a maximum of 1.7 nm, the detection limits of APi-HTOF were ~0.4 and ~4 cm$^{-3}$, respectively. The detection limit of the NAIS decreases with an increase in ion diameter. The detection limit of the NAIS for cluster ions varies from 300 to 30 cm$^{-3}$ at diameters ranging from 0.8 to 2 nm. Thus, the ion concentration measured by the APi-HTOF is more reliable than that measured by the NAIS when the ion concentration is extremely low. As the influence of background noise would diminish when signals increase and the detection efficiency is calculated using data during the whole period, the background noise in the NAIS has limited influence on the calibration we have done above.

Fig. 2 Ion concentrations measured by APi-HTOF and NAIS

**2.3 The simulation of HSO$_4^-$ and C$_3$H$_3$O$_4^-$ concentration**

To quantitatively address the sources and sinks of air ions, we simulate the concentration of ions using the mass balance equation and identify their dominant production and loss pathways. The simulated concentrations are compared with the measured concentrations. HSO$_4^-$ and C$_3$H$_3$O$_4^-$ were chosen as they are one of the most abundant negative ions in different atmospheric environments and their reaction rates with neutral molecules are determined. The production pathways considered in this simulation are the ion-molecular reactions between NO$_3^-$ and neutral molecules H$_2$SO$_4$ and C$_3$H$_4$O$_4$. Previously, it was reported that NO$_3^-$ was formed in the earlier stage of atmospheric ion formation (Beig and Brasseur, 2000; Luts, 1995) and it continues to ionize neutral gaseous that are more acidic than HNO$_3$, such as H$_2$SO$_4$ and C$_3$H$_4$O$_4$ (Eisele, 1989; Tanner and Eisele, 1991). The loss pathways in the atmosphere mainly include condensational loss onto the particles, ion-molecule reactions, and ion-ion recombination with ions of opposite charge. The formation of H$_2$SO$_4$ dimer ions could be an important loss pathway for HSO$_4^-$ (Beck et al., 2022). C$_3$H$_3$O$_4^-$ could further react with H$_2$SO$_4$ and change back to its neutral form C$_3$H$_4$O$_4$ (Beig and Brasseur, 2000). Thus, the following ion-molecule reactions are considered in the dynamic models to simulate the formation of HSO$_4^-$ and C$_3$H$_3$O$_4^-$ in urban Beijing.

$$NO_3^- + H_2SO_4 \xrightarrow{k_1} HSO_4^- + HNO_3 \qquad \text{R.1}$$

$$NO_3^- + C_3H_4O_4 \xrightarrow{k_2} C_3H_3O_4^- + HNO_3 \qquad \text{R.2}$$

$$\mathrm{C_3H_3O_4^- + H_2SO_4 \xrightarrow{k_3} HSO_4^- + C_3H_4O_4}$$ R.3
$$\mathrm{HSO_4^- + H_2SO_4 \xrightarrow{k_4} H_2SO_4 \cdot HSO_4^-}$$ R.4
where $k_1$ to $k_4$ are the rate constants of these reactions, which are $2.32\times10^{-9}$, $2.5\times10^{-9}$, $2.0\times10^{-9}$, and $2.0\times10^{-9}$ cm$^3\cdot$s$^{-}$
$^1$ respectively (Beig and Brasseur, 2000; Lovejoy and Curtius, 2001; Viggiano et al., 1997). The ion-molecule reactions
between HSO$_4^-$ and organic molecules are not considered in this model due to the lack of reaction rate constants.
The mass balance equations for HSO$_4^-$ and C$_3$H$_3$O$_4^-$ can be written as Eq.4 and Eq.5, respectively.
$$\frac{d\mathrm{[HSO_4^-]}}{dt} = k_1\mathrm{[NO_3^-][H_2SO_4]} - k_4\mathrm{[HSO_4^-][H_2SO_4]} - \mathrm{CS[HSO_4^-]} - \alpha n^+\mathrm{[HSO_4^-]}$$ Eq.4
$$\frac{d\mathrm{[C_3H_3O_4^-]}}{dt} = k_2\mathrm{[NO_3^-][C_3H_4O_4]} - k_3\mathrm{[C_3H_3O_4^-][H_2SO_4]} - \mathrm{CS[C_3H_3O_4^-]} - \alpha n^+\mathrm{[C_3H_3O_4^-]}$$ Eq.5
where CS[HSO$_4^-$] and CS[C$_3$H$_3$O$_4^-$] are loss rates caused by the condensational loss onto particles; $\alpha n^+$[HSO$_4^-$] and
$\alpha n^+$[C$_3$H$_3$O$_4^-$] are loss rates caused by the ion-ion recombination between ions of opposite charge; the value of $\alpha$ is
assumed to be $1.6\times10^{-6}$ cm$^3\cdot$s$^{-1}$ (Bates, 1982; Zauner-Wieczorek et al., 2022); $n^+$ is the total concentration of positive
ions and is measured by NAIS. Assuming a steady state, the concentration of HSO$_4^-$ and C$_3$H$_3$O$_4^-$ are calculated as Eq.6
and Eq.7, respectively. HSO$_4^-$ formed through the ion-molecule reaction between C$_3$H$_3$O$_4^-$ and H$_2$SO$_4$ is neglected
considering the low concentration of C$_3$H$_3$O$_4^-$ in Beijing.
$$\mathrm{[HSO_4^-]} = \frac{k_1\mathrm{[NO_3^-][H_2SO_4]}}{\mathrm{CS} + k_3\mathrm{[H_2SO_4]} + \alpha n^+}$$ Eq.6
$$\mathrm{[C_3H_3O_4^-]} = \frac{k_5\mathrm{[NO_3^-][C_3H_4O_4]}}{\mathrm{CS} + k_2\mathrm{[H_2SO_4]} + \alpha n^+}$$ Eq.7
**3. Results and discussion**
**3.1 Negative cluster ion concentration and composition in urban Beijing**
The median concentration of negative cluster ions is 85 cm$^{-3}$ in Beijing during the measurement period, and its 25$^{th}$ and
75$^{th}$ percentiles are 61 cm$^{-3}$ and 112 cm$^{-3}$, respectively. These values are significantly lower than concentrations reported
at typical rural, forest, and marine sites (Fig. 3a). The total concentration of air ions ranges between 14 cm$^{-3}$ and 467 cm$^{-}$
$^3$ in urban Beijing, similar to that in the Indian urban city, Pune, which is also a highly polluted site (Gautam et al., 2017).
Ion concentrations in these two polluted urban cities are so far the lowest, significantly lower than at other urban, rural
and marine, and forest sites, which are mostly above 200 cm$^{-3}$ (Chen et al., 2017; Dos Santos et al., 2015; Hirsikko et
al., 2011; Tammet, 2015a; Tammet, 2015b). It should be noted that although Paris is an urban site, its air is relatively
clean in terms of aerosol concentration compared to the Chinese and Indian polluted urban sites. Similarly, cluster ion

concentrations at some urban sites can also be as high as those at rural and remote sites due to the relatively clean atmosphere (Hirsikko et al., 2011).

Fig. 3 Ion concentration comparison between Beijing and other sites.

The ion concentrations are negatively correlated with CS at both the urban site Beijing and the clean forest site Hyytiälä. In urban Beijing, the median ion concentration decreases from 181 to 51 ions $cm^{-3}$ when CS increases from 0.0032 to 0.12 $s^{-1}$ (Fig. 3b). The negative correlation between the ion concentration and CS exists for all the mobility ranges and is more intensive for ions with mobilities between 1.72 and 0.90 $cm^2 \cdot V^{-1} \cdot s^{-1}$, corresponding to a diameter range of about 1.03-1.44 nm (Fig. S5). A similar negative correlation is also observed in Hyytiälä, where the median ion concentration decreases from 606 to 369 ions $cm^{-3}$ when CS increases from 0.00017 to 0.012 $s^{-1}$. Whereas, the inverse relation between the negative cluster ion concentration and CS is much stronger in Beijing than in Hyytiälä. Combining data from these two sites, a negative correlation can still be seen over a wide CS range from 0.00017 to 0.12 $s^{-1}$. The influence of CS on the positive cluster ions also exists, as positive and negative ions are closely related to each other (Fig. S6). Thus, CS is an important influencing factor for the total cluster ion concentrations in both Beijing and Hyytiälä. This is very likely the reason for the lowest ion concentrations at the two polluted sites (Beijing and Pune, Fig. 3a) having the highest aerosol mass loadings.

In urban Beijing, the estimated lifetime of cluster ions due to the combined effect of condensation loss and ion-ion recombination of cluster ions is 0.02-0.4 min during haze periods ($PM_{2.5} > 75$ μg/m$^3$, average CS = 0.091 $s^{-1}$) and 0.1-1.6 min during clean periods ($PM_{2.5} < 75$ μg/m$^3$, average CS = 0.023 $s^{-1}$) (Fig. S7). The lifetime of cluster ions in Hyytiälä is between 0.8-14 min (average CS = 0.0018 $s^{-1}$), much longer than that in Beijing. Such distinct CS-dependence of total cluster ions in urban Beijing is rarely reported in previous studies despite that the condensational loss of ions on large particles has been widely recognized as important. On the contrary, most of the studies have found that weather conditions and ionizing radiation are the controlling factors for ion concentrations. For example, ion production rates in Hyytiälä were reported to be largely affected by variations in seasonal radiation and wind speed (Chen et al., 2016). This indicates that CS may be the driving factor of cluster ion concentration at highly polluted sites with high aerosol mass loadings and relatively constant ion production rates, while ion production rates may be the driving factor at relatively clean sites where aerosol mass loadings are low and ion production rate varies significantly (Hirsikko et al., 2007).

The composition of negative cluster ions is mainly comprised of inorganic nitrogen-containing ions, inorganic sulfur-containing ions, and organic ions in urban Beijing, accounting for 20-22%, 8-15%, and 37-43%, respectively (Fig. 4 & Fig. S8). $NO_2^-$, $NO_3^-$, and $HNO_3NO_3^-$ are the most abundant among the detected inorganic nitrogen-containing ions. They possibly exist in the forms of $NO_2^- \cdot (H_2O)_n$, $NO_3^- \cdot (H_2O)_n$, and $NO_3^- \cdot HNO_3 \cdot (H_2O)_n$ in the atmosphere (Luts, 1995), with the loosely bounded water molecules being evaporated from the cluster ions when passing through the mass spectrometer. The concentration of $NO_2^-$ and $NO_3^-$ were observed to be well correlated with each other in the atmosphere

of Beijing, and the concentration of $NO_2^-$ is ~20 % of that of $NO_3^-$ (Fig. S9a). This is consistent with the ion chemical models suggesting that $NO_2^-$ and $NO_3^-$ can be stably formed through a series of reactions between primary ions, NO, $NO_2$, and $HNO_3$ in the atmosphere (Beig and Brasseur, 2000; Kawamoto and Ogawa, 1984). $HNO_3NO_3^-$ is subsequently formed by adding an $HNO_3$ molecule to $NO_3^-$. Inorganic sulfur-containing ions are mainly in the form of $HSO_4^-$, $SO_5^-$, and $H_2SO_4HSO_4^-$. The ion $HSO_4^-$ is mainly produced by the ion-molecule reaction between $NO_3^-$ and $H_2SO_4$ which will be discussed hereinafter. $H_2SO_4HSO_4^-$ is formed by the further addition of an $H_2SO_4$ molecule, which is proposed as the first step of ion-induced nucleation (Lovejoy, 2004). $SO_5^-$ is likely to be generated through the reaction between $O_2$ and $SO_3^-$ (Möhler et al., 1992), and its signal variation is similar to that of $HSO_4^-$ (Fig. S9b). Similar compositions and variations of inorganic sulfur-containing ions were also observed at Hyytiälä (Ehn et al., 2010).

Fig. 4 Ion composition in Beijing and Hyytiälä

The negative organic ions are mainly CHO-related or CHON-related organic ions in the form of the adduct with $NO_3^-$ or $HSO_4^-$ (Fig. 4). Only minor fractions (~4%) are in the deprotonated form of $CHO^-$ or $CHON^-$. These indicate that the ionization schematic of organic ions is mainly through ion-molecular reaction with $NO_3^-$ or $HSO_4^-$ in Beijing. As few neutral sulfur-containing organics were observed, it is unlikely that the identified $CHON-HSO_4^-$ is a cluster of sulfur-containing organics and $NO_3^-$, but rather a cluster of nitrogen-containing organics and $HSO_4^-$. Among all negative organic ions, CHON organic ions adducted with $NO_3^-$ ($CHON·NO_3^-$) are the most abundant and account for 56% and 69% during clean and haze periods, respectively. Here the haze periods were identified based on whether $PM_{2.5}$ concentration is higher than 75 $\mu g·m^{-3}$. Though the fraction of $CHON·NO_3^-$ does not change much, their compositions are significantly different between clean and haze periods, mainly influenced by the differences in neutral gaseous molecules. During the haze periods where the average $NO_x$ concentration is 30.1±14.6 ppb, the organic ions are dominated by nitrated phenols (NPs), such as $C_6H_5NO_3·NO_3^-$ and $(C_6H_5NO_3)_2·NO_3^-$. During the clean periods when the average $NO_x$ decreases to 15.0 $\pm$11.5 ppb, a series of organic ion peaks were observed for $m/z$ above 300. The highest peaks in the series include $C_{10}H_{15}NO_{10}·NO_3^-$, $C_{10}H_{14}O_{11}·NO_3^-$, and $C_{10}H_{16}NO_{11}·NO_3^-$, which are possibly the adducts of $NO_3^-$ and highly oxygenated products from monoterpenes under high $NO_x$ conditions (Yan et al., 2020). NPs have high signals in urban Beijing and their properties are different from most other CHON compounds, such as volatilities and charging efficiencies as will be discussed below. Thus, CHON species were divided into $CHON_{NPs}$ and $CHON_{nonNPs}$ groups in the following analysis (Nie et al., 2022).

The negative cluster ion spectra in urban Beijing are significantly different from that in Hyytiälä, possibly due to the high $NO_x$. Contrary to Beijing, CHO-related organic ions are more abundant than CHON-related in Hyytiälä, with fractions of 67% and 33%, respectively. The higher CHON ion fractions in urban Beijing are consistent with previous studies showing that neutral OOMs are composed of more nitrogen-containing species in urban Beijing compared to Hyytiälä due to the high $NO_x$ concentration (Li et al., 2022; Qiao et al., 2021). In addition, the higher $CHON_{NPs}$ ion fractions in urban Beijing are also the result of high anthropogenic emissions of aromatics and high $NO_x$ concentrations

(Cheng et al., 2021). Another difference is that the monoterpene-related OOM dimer ions are abundant in Hyytiälä (Isidorov et al., 1985) but are rarely observed in Beijing. This is also related to the neutral molecules as the formation of neutral OOM dimers would be suppressed under high $NO_x$ conditions (Nie et al., 2022). Moreover, a larger fraction of the organic ions are in the form of adducts with $HSO_4^-$ in Beijing than in Hyytiälä. This is related to the higher $HSO_4^-$/$NO_3^-$ ratio in Beijing (~0.36) compared with Hyytiälä (~0.05). The fraction of $HSO_4^-$ adducted ions would increase during clear days in Hyytiälä but its fraction is still lower than those in Beijing (Bianchi et al., 2017). Despite the different compositions between urban Beijing and forest Hyytiälä, the ion mobility distributions are generally quite similar (Fig. S10). The mode mobility peaks of ions for both sites are ~2 $cm^2 \cdot V^{-1} \cdot s^{-1}$, corresponding to a mobility range dominated by $HSO_4^-$ dimer, trimer, and organic ions. These indicate that similar ion mobility distributions do not necessarily indicate similar ion compositions and direct measurements should be performed.

Comparing the organic ions and neutral molecules measured by the APi-HTOF and the (nitrate-) CI-APi-LTOF, we found that the composition of organic ions is similar to that of neutral molecules after subtracting the reagent ions (Fig. S11). The element number distributions of organic compounds detected in the APi-HTOF and CI-APi-LTOF are similar, in which the overlapping region has C atoms in the range of 3 to 14, O atoms in the range of 1 to 15, and N atoms in the range of 0 to 2 (Fig. S12). There are in total 293 organic species measured both in organic ions and neutral molecules, and in addition, inorganic sulfur-containing ions such as $(H_2SO_4)_{0-3}HSO_4^-$ were measured in both instruments as well. The neutral molecules detected by CI-APi-TOF were identified by subtracting one $NO_3^-$. Such similar elemental distributions and abundant common ions in APi-HTOF and CI-APi-LTOF indicate that naturally charging processes in the atmospheres are quite similar to the $NO_3^-$ charging processes in the CI-APi-TOF, despite that $HSO_4^-$ also has a considerable concentration in the atmosphere and would charge neutral molecules as efficiently as $NO_3^-$. Here the reagent ions such as $NO_3^-$ and $HSO_4^-$ were already subtracted and only the neutral molecular formulas were compared. Organic compounds with carbon atom numbers less than 3 were not considered in our analysis. In addition, most molecules detected only by CI-APi-TOF contain more C atoms of up to 20. This may be because the concentrations and signals of organic ions with such high carbon numbers are too low to be assigned with a certain formula in the APi-HTOF.

**3.2 Formation of negative cluster ions**

To further characterize the formation of negative cluster ions, the influences from the reagent ions (i.e., $NO_3^-$ and $HSO_4^-$) and the neutral molecules were explored in this section. Ions in form of adducts with $NO_3^-$ and $HSO_4^-$ were compared to explore their ionization selectivity towards different compounds. The concentration variations of cluster ions were compared with that of the neutral molecules and the reagent ions to explore their driving factors. Then the charge fractions for different compositions were compared.

**3.2.1 The influence of reagent ions**

The two main reagent ions, $NO_3^-$ and $HSO_4^-$, have different concentrations and variations in urban Beijing (Fig. S13).

Firstly, $NO_3^-$ is about 3 times higher than $HSO_4^-$. During the measurement period, the concentration of $NO_3^-$ is $9.6\pm6.0$
$cm^{-3}$, and that of $HSO_4^-$ is $3.2\pm3.4$ $cm^{-3}$ in urban Beijing. Similarly, $NO_3^-$ is also more abundant than $HSO_4^-$ in Hyytiälä,
indicating that $NO_3^-$ has greater chances to collide with neutral molecules than $HSO_4^-$ in the atmosphere. Secondly, both
$NO_3^-$ and $HSO_4^-$ increase with the increase of total negative cluster ions in urban Beijing, while it is reversed in Hyytiälä.
This illustrates the significant impact of CS on the concentration of cluster ions in urban Beijing, that is, with the decrease
of CS, both the concentrations of reagent ions and product ions increase to different degrees. Moreover, the variation of
$HSO_4^-$ is more significant than that of $NO_3^-$ in urban Beijing. $NO_3^-$ maintains a relatively constant proportion ($15\%\pm5\%$)
in the total negative cluster ions during the whole sampling period in urban Beijing, while the fraction of $HSO_4^-$ varies
significantly due to the strong diurnal variation of $H_2SO_4$. The reason for the relatively constant $NO_3^-$ fraction in total
negative cluster ions needs further investigation, and this phenomenon might offer a chance of combining $NO_3^-$ and
organic ions attached to $NO_3^-$ to predict the neutral molecules.
The compositions of molecules that cluster with $NO_3^-$ and $HSO_4^-$ are generally similar, with $NO_3^-$ clustering with a few
more compounds with high carbon numbers and low oxygen numbers. Fig. 5a shows the carbon atom numbers and
average carbon oxidation states ($\overline{OS}_C$) of molecules that are attached to $NO_3^-$ and $HSO_4^-$. For molecules with less than 8
carbon atoms, molecules attached to $NO_3^-$ and $HSO_4^-$ has similar $\overline{OS}_C$ between -3 and 3. For molecules with more than
8 carbon atoms, molecules with lower $\overline{OS}_C$ (<-0.5) are only attached to $NO_3^-$. The average $\overline{OS}_C$ of the molecules attached
to $NO_3^-$ and $HSO_4^-$ are -0.3 and 0.22, respectively, and the former is close to the molecules detected in CI-APi-LTOF (-
0.45). Similarly, the average O/C ratio for molecules adducting with $NO_3^-$ and neutral molecules detected in CI-APi-
LTOF are 0.85 and 0.82, respectively, slightly lower than the average O/C ratio of molecules adducting with $HSO_4^-$
(1.03). These indicate that the selectivity of $HSO_4^-$ and $NO_3^-$ towards organic molecules are similar, with only slight
differences for compounds with high carbon numbers and low oxygen numbers. The slight difference may be partly
because the concentrations of these adducts of these compounds and $HSO_4^-$ are too low to be accurately distinguished.
Overall, O/C ratio of compounds adducted with both $NO_3^-$ and $HSO_4^-$ are both significantly higher than those detected
by the (iodide)CI-APi-TOF and (oxygen)CI-APi-TOF (Li et al., 2021; Riva et al., 2019), indicating their higher
selectivity for oxidized compounds, which is consistent with previous theoretical simulations (Nadykto et al., 2018). In
addition, when only CHO and $CHON_{nonNPs}$ molecules clustering with both $NO_3^-$ and $HSO_4^-$ are considered, their volatility
distributions are very similar when clustering with $NO_3^-$ or $HSO_4^-$, which further indicates that the selectivity difference
between $NO_3^-$ and $HSO_4^-$ towards organic molecules is very small. (Fig. 5b & Fig. S14).

417                        Fig. 5 The species of organic ions adducting with $NO_3^-$ and $HSO_4^-$

For the 88 species that form clusters with both $NO_3^-$ and $HSO_4^-$, the ratio of the two ionization forms ($M\cdot HSO_4^-/M\cdot NO_3^-$)
is positively related to the ratio of reagent ions ($HSO_4^-/NO_3^-$). In addition, $HSO_4^-$ forms clusters with CHO and
$CHON_{nonNPs}$ more efficiently, while $NO_3^-$ forms clusters with $CHON_{NPs}$ more efficiently, despite the above-mentioned
similarity in their selectivity towards different organic molecules. There are an overall of 33 CHO, 40 $CHON_{nonNPs}$, and
15 $CHON_{NPs}$ species that both form ions with $NO_3^-$ and $HSO_4^-$. The average ratio of $HSO_4^-/NO_3^-$ is 0.36, and the relative
ratio of $M \cdot HSO_4^-/M \cdot NO_3^-$ are 0.74, 0.73, and 0.23 for CHO, $CHON_{nonNPs}$, and $CHON_{NPs}$ species, respectively. Generally,
$M \cdot HSO_4^-/M \cdot NO_3^-$ ratio is positively related to $HSO_4^-/NO_3^-$ ratio, and the dots are close to the 1:1 line as shown in Fig. 6.
Specifically, the dots for $CHON_{NPs}$ are mostly lower than 1:1 line, especially under high $H_2SO_4$ concentration. Similar
decreasing trends were also observed for the $M \cdot HSO_4^-/M \cdot NO_3^-$ ratio for CHO and $CHON_{nonNPs}$. This is possibly due to
the competition between different neutral compounds for the reagent ions, as the atmospheric concentration of ions is up
to 7 orders of magnitude lower than neutral molecules. For example, gaseous $H_2SO_4$ may compete with the organic
molecules to react with $HSO_4^-$. As a highly electronegative substance, $H_2SO_4$ prefers to be negatively charged and form
strongly bounded clusters with alkaline substances such as $HSO_4^-$ or amines in the atmosphere, thus $HSO_4^-$ would prefer
to cluster with $H_2SO_4$ during daytime and its efficiency of clustering with organic molecules drops as a result. During
the nighttime, when the competition from $H_2SO_4$ weakens, the formation of $M \cdot HSO_4^-$ increases.

433            Fig. 6 The ratio between organic ions adducting with $NO_3^-$ and $HSO_4^-$

**3.2.2 The influence of neutral molecules**

The diurnal cycles of negative cluster ions are mainly determined by their corresponding neutral molecules, and
exceptions were found for $CHON_{NPs}$-related ions, the diurnal cycles of which are more related to the reagent ions (Fig.
7). The reagent ions and their inorganic clusters ($(HNO_3)_{0-2}NO_3^-$ and $(H_2SO_4)_{0-2}HSO_4^-$) both reach their maximum at
13:00 but show different diurnal patterns. $(H_2SO_4)_{0-2}HSO_4^-$ has a single peak pattern that resembled neutral $H_2SO_4$, while
$(HNO_3)_{0-2}NO_3^-$ has a three-peak pattern that matched the total concentration of negative cluster ions. For the organic
ions, the CHO and $CHON_{nonNPs}$-related ions both peak in the afternoon, which resemble the diurnal pattern of
corresponding neutral molecules. However, the diurnal patterns of $CHON_{NPs}$-related ions and neutral $CHON_{NPs}$ show
opposite variations, and that of $CHON_{NPs}$-related ions is closer to the diurnal variation of $NO_3^-$. As $CHON_{NPs}$-related
ions reach their maximum at nighttime, neutral $CHON_{NPs}$ reach their maximum at both noon and evening, indicating
that the formation of $CHON_{NPs}$-related ions may be different from other organic ions. Besides, although the diurnal
variations of CHO and $CHON_{nonNPs}$-related ions roughly follow those of neutral molecules, it was found that their peak
times coincided with that of $(HNO_3)_{0-2}NO_3^-$ at 13:00 and 17:00. Differently, the diurnal patterns of CHO and
$CHON_{nonNPs}$-related ions tended to peak during nighttime in Hyytiälä (Bianchi et al., 2017). This could be due to the
different diurnal patterns of neutral molecules as those of the reagent ions are similar between Beijing and Hyytiälä.

449            Fig. 7 The diurnal patterns of negative cluster ions, neutral molecules, and charge fractions

Consistently, the cluster ion concentrations were found to be positively correlated with that of neutral molecules under
similar CS for CHO and $CHON_{nonNPs}$ species during the whole sampling period. As shown in Fig. 8, generally, the
concentrations of all organic ions decrease with increasing CS as indicated in Section 3.1. When CS is larger than $0.01 s^-$
$^1$, the concentrations of CHO and $CHON_{nonNPs}$-related ions rise with their corresponding neutral gas concentration under

similar CS levels, while that of CHON$_{NPs}$-related ions remains constant or slightly declines. When CS is smaller than 0.01 s$^{-1}$, which is usually accompanied by low concentrations of neutral molecules, the relationships between organic ions and the corresponding neutral molecules are not obvious, this may be due to a combined effect of neutral molecules and the reagent ions.

Fig. 8 The variation of organic ions with neutral molecules under different CS

The charge fraction between organic ions and neutral molecules reflects a charging competition between different neutral compounds. Here, we calculate charge fraction as the proportion between the concentrations of cluster ions and their corresponding neutral molecules. As shown in Fig. 7c, different organic molecules have similar diurnal variations of charge fractions, which are highest at 5:00 and lowest at 15:00. While the HSO$_4^-$/H$_2$SO$_4$ ratio has the opposite pattern and is highest during the daytime. The opposite diurnal patterns between organic ions and HSO$_4^-$ indicate that H$_2$SO$_4$ and organic molecules would compete for being charged. The median charge fractions of CHO and CHON$_{nonNPs}$ molecules with NO$_3^-$ and HSO$_4^-$ are close to each other and range from 2×10$^{-7}$ to 2×10$^{-6}$, while the charge fractions of CHON$_{NPs}$-related ions are much lower than these organic ions. This suggests that CHON$_{NPs}$ are less likely to form cluster ions with NO$_3^-$ and HSO$_4^-$ than CHO and CHON$_{nonNPs}$. As the charge fraction for organic ions attached to NO$_3^-$ and HSO$_4^-$ has similar diurnal patterns, it is deduced that the diurnal variation of charge fractions is mainly driven by the variation of neutral molecules instead of reagent ions.

For CHO and CHON$_{nonNPs}$, the charge fractions between organic ions and neutral molecules generally increase with the carbon atom numbers and $\overline{OS}_C$ of the molecules. As shown in Fig. 9, the average charge fractions of different organic ions vary between 3×10$^{-8}$ and 8×10$^{-6}$, and species with larger carbon atom numbers and higher oxidation states tend to have higher charge fractions, either when clustering with NO$_3^-$ or HSO$_4^-$ (Fig. S15). The charge fraction difference between molecules with different carbon atom numbers and $\overline{OS}_C$ is a reflection of the differences in reaction rates between reagent ions and the neutral molecules or the stabilities of the cluster ions. In summary, through the comprehensive quantitative analysis of atmospheric cluster ions, reagent ions, and neutral molecules, we can roughly predict the composition and concentration variations of atmospheric negative cluster ions from measured neutral molecules and CS, as indicated by Fig. 8. However, as the actual charging fractions for different species differ in a wide range as shown in Fig. 9, the prediction of actual concentrations of atmospheric ions would require more work.

Fig. 9 The influence of molecular characteristics on the charge fraction

**3.3 Quantification of the sources and sinks of representative negative cluster ions**

We found that the ionization of H$_2$SO$_4$ and C$_3$H$_4$O$_4$ by NO$_3^-$ and condensational loss to particles are the main formation and loss pathways for HSO$_4^-$ and C$_3$H$_3$O$_4^-$ respectively. As shown in Fig. 10, the simulated concentrations of both HSO$_4^-$ and C$_3$H$_3$O$_4^-$ by Eqs.6-7 have a good consistency with the measured concentrations, indicating that the considered production and loss pathways can reproduce the measured ion concentration in urban Beijing. The ionization of H$_2$SO$_4$

and $C_3H_4O_4$ by $NO_3^-$ is the main production pathway. For both $HSO_4^-$ and $C_3H_3O_4^-$, the condensational loss is the main loss pathway, contributing to more than 70% of the total loss rate, which explains why CS is a driving factor for the concentration of cluster ions in urban Beijing. Ion-ion recombination process only accounts for 6% and 8% of the total loss rates for $HSO_4^-$ and $C_3H_3O_4^-$, which is due to the low ion concentrations in the polluted urban atmosphere. Notably, the ion-molecular reactions between the ions with $H_2SO_4$ could contribute to 19% of the loss of both $HSO_4^-$ and $C_3H_3O_4^-$. The transformation from $C_3H_3O_4^-$ back to $C_3H_4O_4$ indicates that the conversion of ions back to neutral molecules is also significant.

Fig. 10 Simulation of the sources and sinks of $HSO_4^-$ and $C_3H_3O_4^-$.

**4 Conclusions**

We quantified the composition-resolved ion concentrations in the atmosphere by combining mass spectrometry and electrical mobility measurements. The absolute transmission efficiency in the mass spectrometry APi-HTOF obtained with the ion mobility spectrometer (NAIS in this study) agreed well with the relative transmission efficiency obtained in the chemical ionization mode of the APi-HTOF. The calibrated ion concentrations with different *m/z* ranges agreed well with the concentration of ions with corresponding mobility ranges of the ion mobility spectrometer. These indicate that an APi-HTOF can be well calibrated by running an APi-HTOF and an ion mobility spectrometer side by side in ambient measurements. Furthermore, we propose that the transmission efficiency of a CI-APi-TOF can also be obtained through *in-situ* comparison with an ion mobility spectrometer, as long as the X-ray and voltages of a chemical ionization inlet are turned off and atmospheric ions are directly measured.

The cluster ion concentrations and composition in urban Beijing are largely affected by the high condensation sink and nitrogen oxides. Firstly, ion concentration decreased significantly with an increasing CS for cluster ions in all the mobility ranges. Median cluster ion concentrations in urban Beijing were only ~85 cm$^{-3}$, much lower than those reported at clean and rural sites due to the high CS in urban Beijing. Due to high concentrations of nitrogen oxides, the organic ion compositions were composed of more nitrogen-containing species in urban Beijing than those in Hyytiälä. This is consistent with the higher neutral nitrogen-containing molecular fractions in urban Beijing as previously reported.

The formation of cluster ions was analyzed by comparing their concentrations, volatility distributions, and charge fractions with CS, reagent ions, and their corresponding neutral molecules. The organic cluster ions are mainly in the form of adducts with $NO_3^-$ or $HSO_4^-$. The ratio of the two ionization forms ($M \cdot HSO_4^-/M \cdot NO_3^-$) is positively related to the ratio of reagent ions ($HSO_4^-/NO_3^-$). Although the molecules clustered with $HSO_4^-$ and $NO_3^-$ are similar in composition, we found that $M \cdot HSO_4^-$ formed more efficiently for CHO and CHON$_{nonNPs}$, while $M \cdot NO_3^-$ formed more efficiently with CHON$_{NPs}$ when targeting the compounds that can be ionized by both $HSO_4^-$ and $NO_3^-$. The concentrations of organic ions are positively correlated with that of neutral molecules, resulting in their similar diurnal cycles. However, an

exception was found for $CHON_{NPs}$, the concentration of which is also significantly influenced by the reagent ions $NO_3^-$. The charge fractions tend to be higher for organic molecules with higher molecular weight and oxidation state. We also observed a charging competition between different neutral organic and inorganic compounds, such as the competition between $H_2SO_4$ and organic ions. Through this quantitative analysis, it is possible to infer the variation of atmospheric negative cluster ions from the measured neutral molecules and CS in the urban atmosphere. The formation pathway of $HSO_4^-$ and $C_3H_3O_4^-$ were well characterized by the ionization of $H_2SO_4$ and $C_3H_4O_4$ by $NO_3^-$, and their loss processes are dominated by condensation loss, with minor contributions from ion-molecular reactions and ion-ion recombination.

Inspired by the high dependence of charge fraction on the molecule species, we can take the ambient atmosphere as a natural ion-molecule reaction chamber, and the charge fraction observed for different neutral molecules may provide some insights into the charging efficiency of organic species measured with the reagent ions of $NO_3^-$ and $HSO_4^-$ in the chemical ionization mass spectrometers and help better quantify them.

**Data availability**

The detection efficiency of APi-TOF, the time series of total negative cluster ions and CS in urban Beijing, and the average spectrums of negative clusters ions measured by APi-TOF during haze and clean periods in urban Beijing are available from https://zenodo.org/record/7791785.

**Author contributions**

RY and JJ designed the study. RY, XL, CY, RC, YZ, JK, NS, and JL participated in data collection and performed the data analysis. RY and XL prepared the first version of the manuscript with contributions from all co-authors. All authors approved the final version of the manuscript.

**Competing interests**

The authors declare that they have no conflict of interest.

**Acknowledgments**

This research has been supported by National Natural Science Foundation of China (grant no. 22188102 and 22106083), Samsung PM$_{2.5}$ SRP, the Academy of Finland (grant no. 337549, 302958, 337549, 1325656, 316114, 325647, and 332547), and European Research Council (grant no. 742206). The authors gratefully acknowledge the support of the

research teams in AHL/BUCT laboratory and SMEAR II station.

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

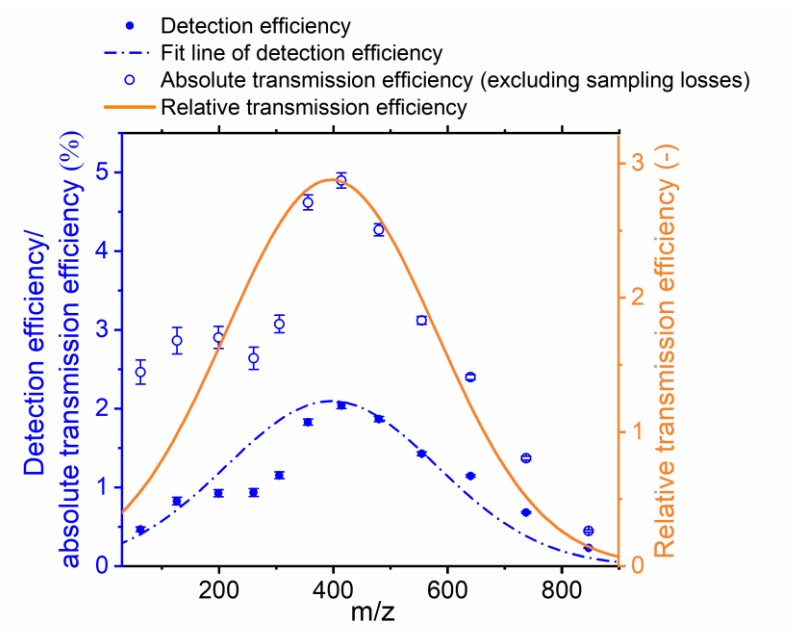

782

**Figure 1.** The detection efficiency and absolute transmission efficiency of APi-HTOF determined through the *in-situ* comparison with NAIS and the relative transmission efficiency of APi-HTOF determined through the depletion method. The relative transmission efficiency is obtained by dividing the detection efficiency by the sampling efficiency of cluster ions. The voltage settings of APi-HTOF remain the same during the experiment.

787

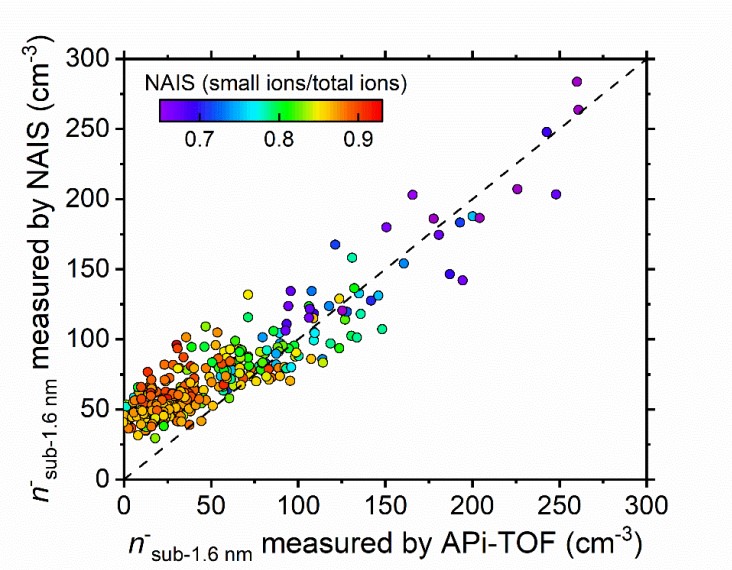

788

**Figure 2.** The total concentration of negative cluster ions ($n^-_{sub\text{-}1.6\,nm}$) measured by APi-HTOF is well correlated with that measured by NAIS. Colors represent the ratio of negative ions smaller than 1.3 nm ($n^-_{sub\text{-}1.3\,nm}$) to negative cluster ions ($n^-_{sub\text{-}1.6\,nm}$).

793

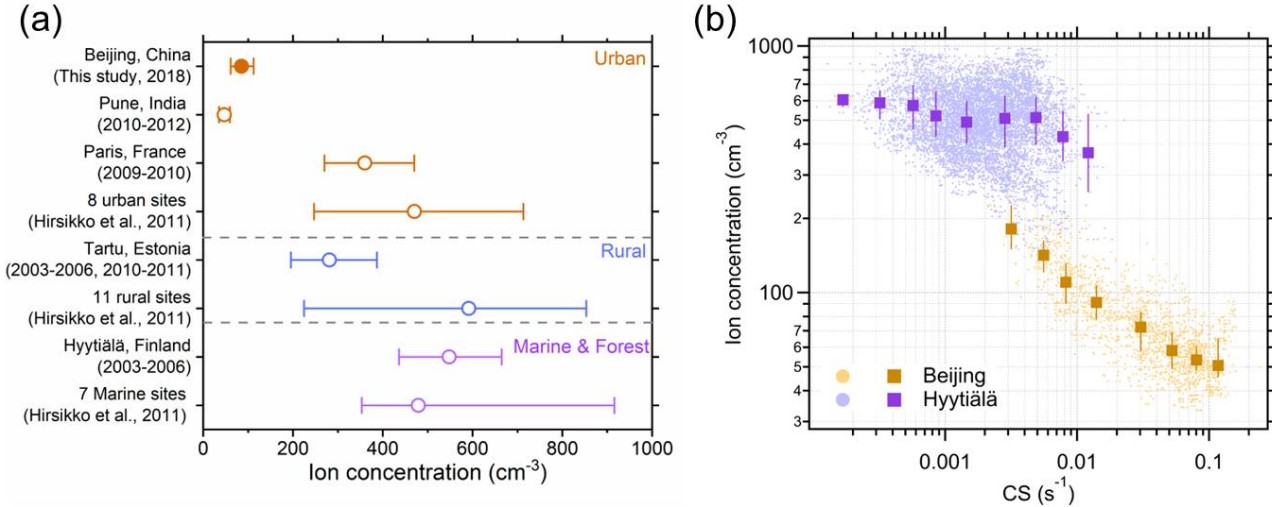

794

**Figure 3.** (a) Negative cluster ions observed in global urban, rural, forest, and marine sites. The circles represent the median values, and the error bars represent the 25%-75% ranges. Ion mobility ranges for different sites were between sub-1.6 and sub-2 nm. (b) The decreasing trends of the negative cluster ion concentrations with the increase of CS were observed in both Beijing and Hyytiälä. The markers and error bars are median values and 25%-75% ranges, respectively. For comparison, NAIS data were used for both urban Beijing and the clean forest site Hyytiälä.

801

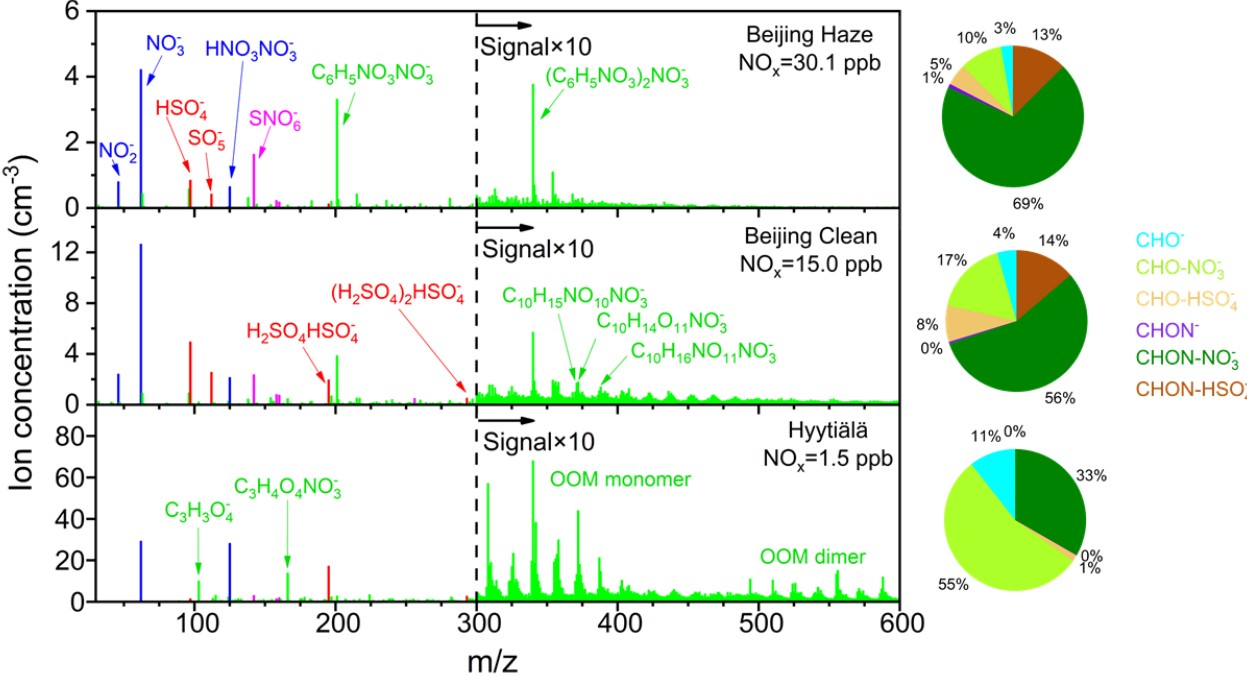

802

**Figure 4.** The negative cluster ion compositions in Beijing and Hyytiälä. The bar plots were average mass spectra of negative cluster ions during clean and haze periods in Beijing and Hyytiälä measured by APi-HTOF. Signals for *m/z*>300 were multiplied by 10 times in Beijing for clearer views. Haze periods were identified based on whether $PM_{2.5}$ concentration is higher than 75 $\mu g \cdot m^{-3}$. The average $NO_x$ concentrations for the three situations are 30.1, 15.0, and 1.5 ppb, respectively. The pie charts were species distribution of organic ions, including the deprotonated form, adducts to $NO_3^-$ and $HSO_4^-$ forms of CHO and CHON species.

809

810

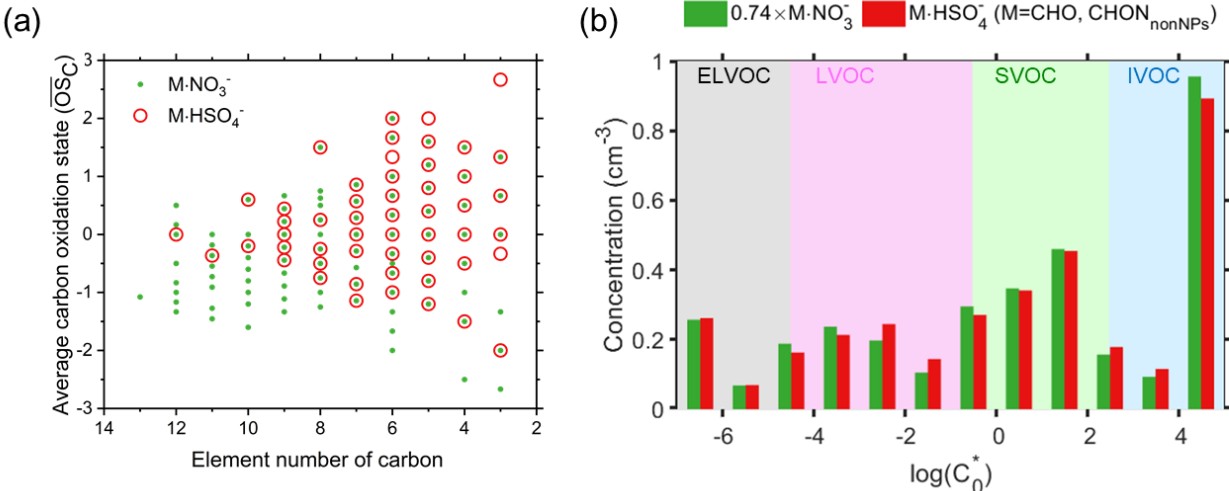

811

**Figure 5.** (a) The carbon atom number and average carbon oxidation state ($\overline{OS}_C$) of molecules that adduct to $NO_3^-$ and $HSO_4^-$ respectively. (b) The volatility distribution of CHO and $CHON_{nonNPs}$ species in the forms of adducts with both $NO_3^-$ and $HSO_4^-$. Only species that attached to both $NO_3^-$ and $HSO_4^-$ were considered for calculating the volatility distribution. The volatilities were calculated using the method of Qiao et al. (2021).

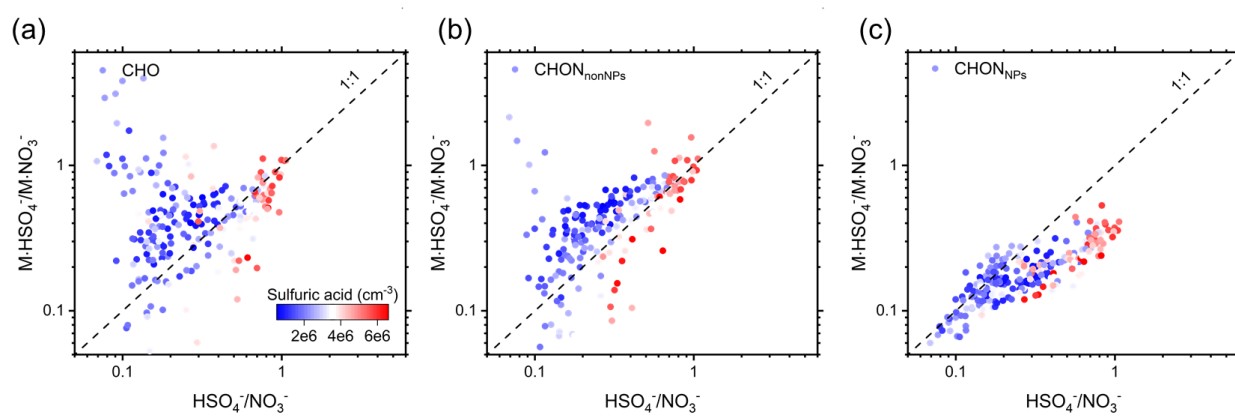



**Figure 6.** The ratio of $CHO \cdot HSO_4^-/CHO \cdot NO_3^-$ (a), $CHON_{nonNPs} \cdot HSO_4^-/CHON_{nonNPs} \cdot NO_3^-$ (b), and $CHON_{NPs} \cdot HSO_4^-$
$/CHON_{NPs} \cdot NO_3^-$ (c) as a function of the ratio of $HSO_4^-/NO_3^-$. The color represents the concentration of sulfuric acid.



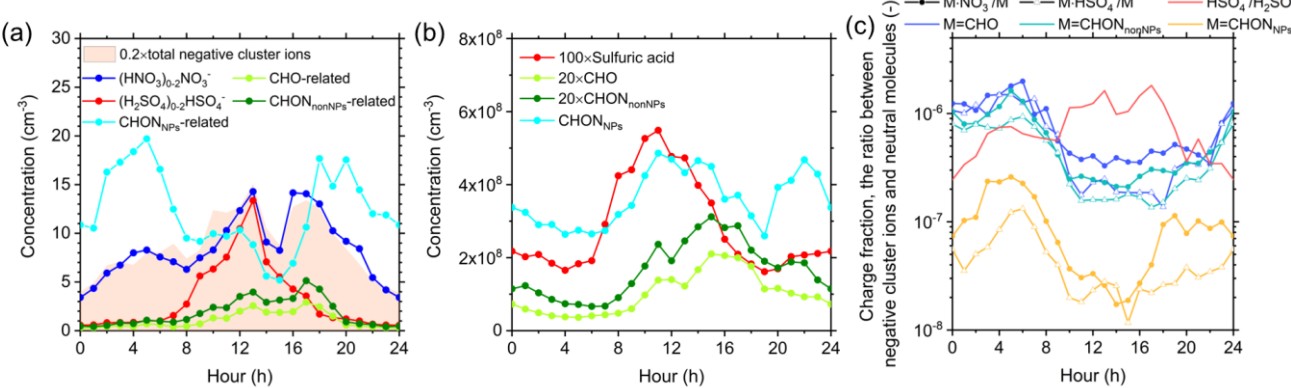


**Figure 7.** Diurnal cycles of negative cluster ions, corresponding neutral molecules, and the charge fractions between
them in urban Beijing. (a) The diurnal cycles of reagent ions and their inorganic clusters, together with those of organic
ions. The organic ions in the forms of adducts with $NO_3^-$ and $HSO_4^-$ were summed into CHO-, $CHON_{nonNPs}$-, and
$CHON_{NPs}$-related ions. (b) The diurnal cycles of sulfuric acid and the corresponding neutral molecules of organic ions
measured by the (nitrate-) CI-APi-LTOF. The concentrations of sulfuric acid, CHO, and $CHON_{nonNPs}$ were multiplied by
a factor of 100, 20, and 20, respectively, to match that of $CHON_{NPs}$ in the figure. (c) The diurnal cycles of charge fractions
between negative cluster ions and their corresponding neutral molecules. The charge fractions for CHO, $CHON_{nonNPs}$,
and $CHON_{NPs}$ in the form of adducts with $NO_3^-$ and $HSO_4^-$ were calculated and compared with the $HSO_4^-/H_2SO_4$ ratio.


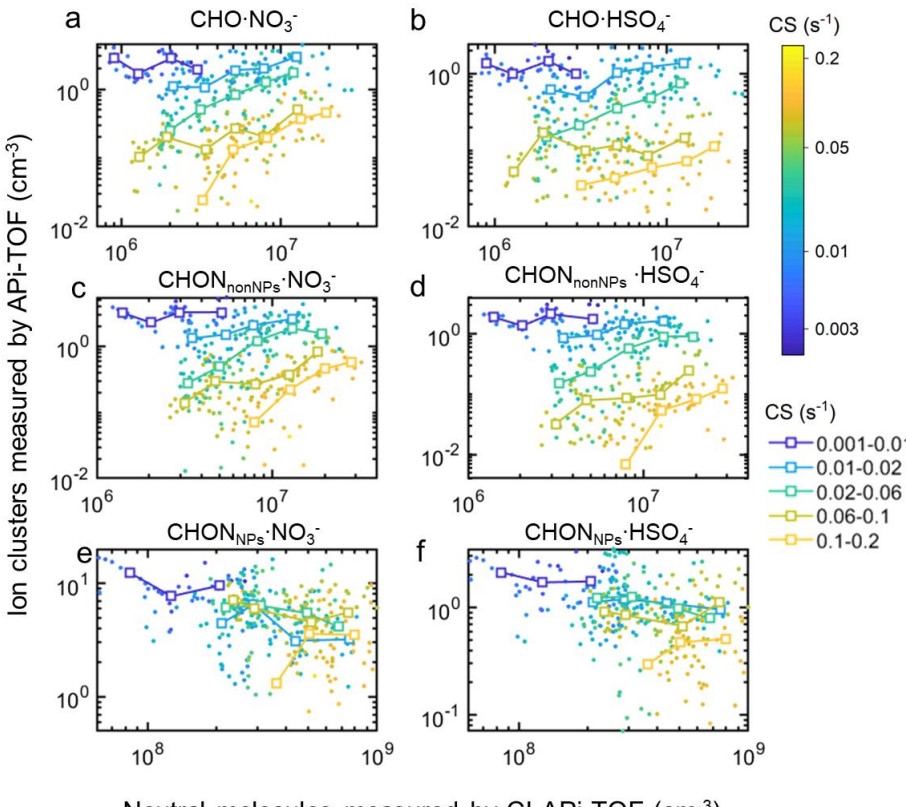


**Figure 8.** Scatter plots of the concentrations of organic ions measured by APi-TOF and their corresponding neutral molecules measured by CI-APi-TOF in urban Beijing, colored by CS. The cluster ions were $CHO \cdot NO_3^-$ (a), $CHO \cdot HSO_4^-$ (b), $CHON_{nonNPs} \cdot NO_3^-$ (c), $CHON_{nonNPs} \cdot HSO_4^-$ (d), $CHON_{NPs} \cdot NO_3^-$ (e), and $CHON_{NPs} \cdot HSO_4^-$ (f). The neutral molecules were CHO (a and b), $CHON_{nonNPs}$ (c and d), and $CHON_{NPs}$ (e and f), respectively. The data was divided into 5 groups based on CS in the range of 0.001-0.01, 0.01-0.02, 0.02-0.06, 0.06-0.1, and 0.1-0.2 $s^{-1}$, respectively.


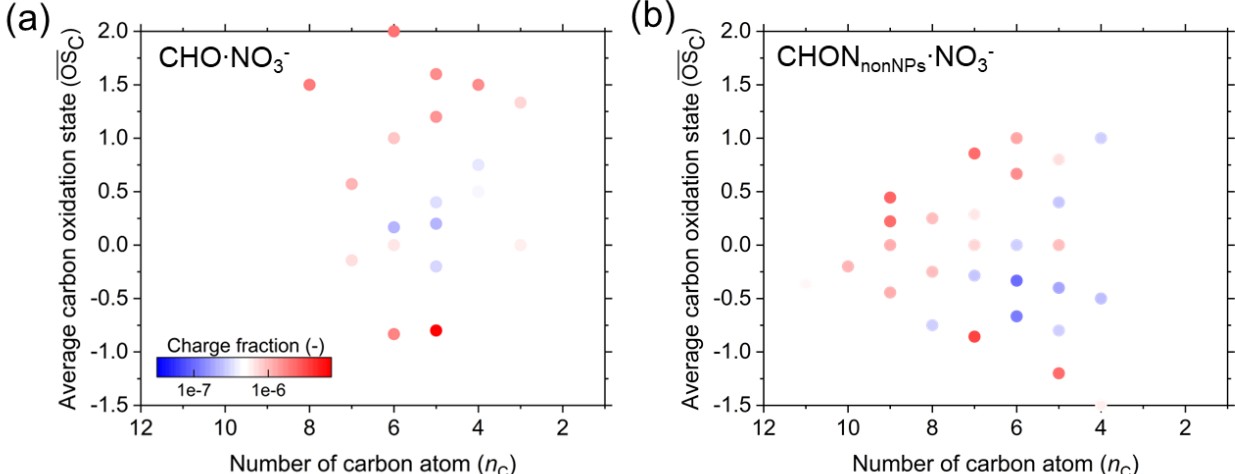

**Figure 9.** The carbon atom number and average carbon oxidation state ($\overline{OS}_C$) of molecules in CHO·NO$_3^-$ (a) and
CHON$_{nonNPs}$·NO$_3^-$ (b). The color represents their charge fraction during the measurement periods.

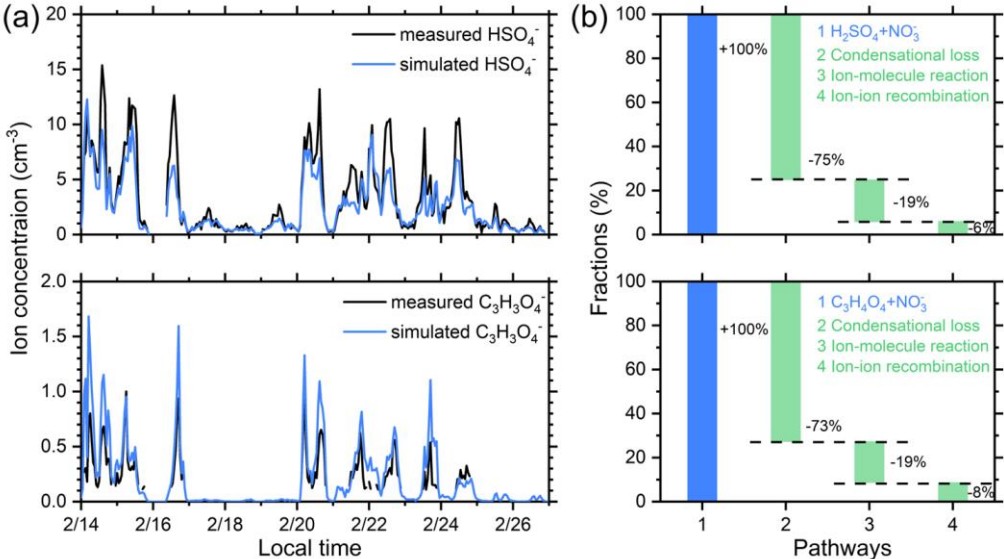


**Figure 10.** (a) The measured and simulated concentrations of $HSO_4^-$ and $C_3H_3O_4^-$ in Beijing. The simulation is performed according to dynamic models as described in Section 2.3 and four formation and loss pathways were considered. (b) The fractions of the formation or loss rates of $HSO_4^-$ and $C_3H_3O_4^-$ contributed by different pathways. The average values of the results were shown here.

853