# Peer review of "Revealing the sources and sinks of negative cluster ions in an urban environment through quantitative analysis"

_EGUsphere, 2022_

## Author Response (AR1)

**Responses to Reviewers' Comments on Manuscript egusphere-2022-1108**

(Characteristics of Negative Cluster Ions in an Urban Environment)

We are grateful for the reviewers' comments and we feel that our responses to these will greatly improve this manuscript. We have addressed the comments in the following paragraphs and made corresponding changes in the revised manuscript. Comments are shown as *blue italic text* followed by our responses. Changes are highlighted in the revised manuscript and shown as "quoted underlined text" in our responses.

**Reviewer #1:**

The manuscript by Yin et al. describes measurements of atmospheric ions in Beijing. Notably, they quantify the concentrations of ions of specific composition through a rigorous calibration process. For select ions, a steady-state analysis is applied to show how the time series of these specific ions are well represented by known chemical and physical processes and that condensational loss represents the largest ion loss term. Ion composition is compared to the composition of neutral gases measured by a nitrate chemical ionization mass spectrometer (CIMS) and similar to past findings, it is found that there is generally good agreement in the types of organic molecules identified as ambient ions and as neutral gases albeit with different intensities. These measurements from an urban environment are also compared to measurements from the boreal forest with similarities and differences discussed.

In my opinion, the most important contribution of this manuscript is the push towards making quantitative measurements of the chemically resolved composition of ambient ions and in using these quantitative measurements to evaluate our understanding of ion sources and sinks. Quantifying specific ions is a challenging undertaking that few have pursued in recent years (specifically when using time-of-flight mass spectrometers). Although some would argue that the science to effort pay off of such measurements may be minor, I think that there are some important questions that could be addressed if such measurements became more routine. The other aspects of the manuscript (discussion of composition, comparison to other environments, etc.) are only superficially explored and require a more comprehensive analysis before they would provide general insight into atmospheric composition and chemistry. In short, I think this manuscript contributes a technical advancement to the field and thus may be more appropriate for a different journal or as a technical note. In my opinion, to be suitable for publication in a more general journal, more detailed analysis regarding composition is necessary and the scientific motivation and insights of the analysis need to be clearly established in the manuscript.

**Response:** We appreciate the reviewer for the affirmative comments on our work of ion quantification. As suggested by the reviewer, we have added more detailed discussions on the ion compositions, sources, and sinks for a deeper scientific understanding of ion formation in the revised manuscript, including (1) the charging capacity of  $NO_3^-$  and  $HSO_4^-$  in the ambient atmosphere was determined by analyzing ions concentrations as well as organic ion compositions. The ratio of the two ionization forms (M·HSO4-/M·NO3-) was positively related to the ratio of reagent ions (HSO4-/NO3-). Although the molecules clustered with HSO4- and ·NO3- are similar in composition, we found that M·HSO4- formed more efficiently for CHO and CHONnonNPs, while M·NO3- formed more efficiently with CHONNPs when targeting the compounds that can be ionized by both HSO4- and ·NO3-. (2) the relationships between ions and neutral molecules were compared. We found that most organic ions were positively correlated with neutral molecules, resulting in their similar diurnal cycles. For these compounds, the ion concentrations can be roughly estimated through the neutral molecular concentrations and CS. However, an exception was found for CHONNPs, the concentration of which is also significantly influenced by the reagent ions  $NO_3^-$ .

charging competition between different neutral organic and inorganic compounds, such as the competition between  $H_2SO_4$ and organic ions. We believe these analyses will largely promote the understanding of the formation of atmospheric ions and formulate important implications on the ionization chemistry of  $NO_3^-$  and  $HSO_4^-$ .

(1) Sect 2.1: At least a basic overview of the measurements at the boreal forest site needs to be provided so that the reader can understand and correctly interpret the results presented. Given that the measurements have been previously published, it is not necessary to go into a lot of detail, but the basics should be provided so the manuscript stands on its own. These include the dates of the measurements, a brief description of the inlet, how (if?) the transmission efficiency was determined, and any extenuating/unusual circumstances that would influence the measurements and the conclusions drawn here.

**Response:** Thanks for the suggestions! We have added detailed information about the measurement of cluster ions in Hyytiälä including the measurement periods and the sampling settings in the revised manuscript as below,

**Table S1 in SI**

Table S1. The measurement period and time resolution for instruments at AHL/BUCT and SMEAR II stations.

| Site, type                 | Location          | Instrument  | Time resolution | Measurement period    |
|----------------------------|-------------------|-------------|-----------------|-----------------------|
| AHL/BUCT,
urban         | Beijing, China    | APi-HTOF    | 1 h             | Feb 14-Feb 27, 2018   |
|                            |                   | CI-APi-LTOF | 5 min           | Jan 23-April 14, 2018 |
|                            |                   | NAIS        | 4.5 min         | Jan 12-Dec 31, 2018   |
| SMEAR II,
boreal forest | Hyytiälä, Finland | APi-HTOF    | 1 h             | April 7-June 8. 2013  |
|                            |                   | NAIS        | 4.5 min         | Jan 1-Dec 31 2013     |

[Line 124-126] "Besides, the measurements at a boreal forest station SMEAR II (System for Measurement of forest Ecosystem and Atmospheric Relationships) located at Hyytiälä, Finland were used for comparison. Details of the instruments and their measurement periods in these two stations are given in Table S1."

[Line 127-129] "The atmospheric negative cluster ion compositions were measured with an atmospheric pressure interface time-of-flight high-resolution mass spectrometers (APi-HTOF) from Feb 14 to Feb 27, 2018, in Beijing, and from April 7 to June 8, 2013 in Hyytiälä (Yan et al., 2018)."

[Line 138-140] "In Hyytiälä, ions were sampled through a core sampling method. It has a wide tube outside and a short coaxial tube inside to minimize the sampling losses with a total flow rate of 5 LPM."

[Line 157-159] "The ion mobility distribution was measured by a NAIS (NAIS, Airel Ltd.) ..... from Jan 1st to Dec 31st, 2013 in Hyytiälä (Kontkanen et al., 2013)."

(2) Sect 2.1: The dates of the measurements are unclear to me. On line 123, it is stated that the study ran from Jan 14th to Sept 16th, 2018, while on line 126 it states that the API measurements are from Feb 14th to Feb 27th, 2018. Were API measurements only made over this ~2 week period? If so, that needs to be clarified throughout the text. Were the nitrate CIMS measurements made concurrently with the API measurements or are they from a different time period? If a different time period, when exactly was that period and how would that influence the comparison between the API and the CIMS? As it reads now, the manuscript implies the measurements are simultaneous. Further information regarding the nitrate CIMS

**inlet, sampling protocol, and calibration frequency also need to be given so the reader can judge the likelihood of potential artefacts etc.**

**Response:** The sampling period of APi-TOF measurement was from Feb 14th to Feb 27th. The measurement period only lasts two weeks because the APi inlet is easy to be blocked by aerosols, dust, or catkin in polluted areas and ion clusters would be hard to be detected, similar to the study of Yao et al. (2018).

A CI-APi-TOF and an APi-TOF were operated side by side to acquire information on the neutral molecules and negative cluster ions simultaneously.

The calibrations, including the sensitivity calibration for  $H_2SO_4$  and the transmission efficiency calibrations for the mass detectors, were calibrated at the end of the campaign.

Apart from the modification listed in question 1, we have revised the manuscript as below,

[Line 144-148] "It was operated side by side with the APi-TOF to achieve simultaneous measurements of neutral molecules and cluster ions.  $NO_3^-$  and its adducts (HNO3)1-2NO3- were used as the reagent ions to chemically ionize the gas molecules with a laminar flow ionization source mounted in front of the APi-LTOF (Eisele and Tanner, 1993). Nitric acid is volatilized and carried by a total of 20 LPM sheath flow, then it is exposed to soft X-ray to produce  $NO_3^-$  and its adducts (HNO3)1-2NO3-."

[Line 151-154] "The sensitivity for  $H_2SO_4$  was calibrated via the controlled reaction between  $SO_2$  and OH at the end of the campaign (Kürten et al., 2012; Li et al., 2019). The relative mass-dependent transmission efficiency of the CI-APi-LTOF was calibrated using the depletion method (Heinritzi et al., 2016)."

[Line 158-159] "In both stations, the measurement periods of NAIS covered the measurement periods of APi-TOF."

**(3) Sect. 2.1: Please provide more details on the inlet for the NAIS and how the calibration was performed. These details, particularly regarding calibration and inlet colocation/design, are critical for evaluating the API quantification.**

**Response:** A 1.8 m long 4 cm copper tube is used to sample ambient ions with a flow rate of 54 LPM to reduce the losses of ions during sampling. NAIS was calibrated through the method described by Wagner et al. (2016) and both inversion kernel and sampling losses were considered in the data inversion. We added the information to the revised manuscript as below,

[Line 167-171] "In urban Beijing, ambient air is drawn into the system through a 1.8 m long copper tube with a diameter of 4 cm, positioned at the window. The sample flow rate is maintained at 54 LPM. The inversion kernel of NAIS was calibrated based on the method described by Wagner et al. (2016) and sampling losses were further considered in the data inversion. The results of NAIS were calibrated considering both the transfer functions and sampling losses (Mirme and Mirme, 2013; Wagner et al., 2016)."

(4) Sect. 3.2: While this section provides description of the measurements, the scientific motivation of the analysis as well as how the results can be used to further broaden our understanding of atmospheric composition/chemistry are unclear to me. The comparison between Beijing and Hyytiälä is rather superficial, just presenting averages. There is no discussion on how seasonality, day-night differences, transport, etc. would influence the results and the comparisons. The finding that the two locations differ in composition is unsurprising and doesn't provide new insights at the given level of analysis. I suggest either removing this analysis or significantly expanding it to consider aspects such as how the possible HOM precursors differ and what that implies about HOM molecules that are often important for aerosol growth. Further specific minor comments about this section are listed below.

**Response:** Thanks for the suggestion. To gain deeper scientific insights into the cluster ion characteristics, we reorganized the contents and added more analysis on the cluster formation processes. For section 3.2, we cut down the comparisons between Beijing and Hyytiälä and only mention data from Hyytiälä as a reference when needed. Now Section 3.2 mainly focuses on the formation of negative cluster ions from the view of both reagent ions and neutral molecules. The formation of cluster ions was analyzed by comparing their concentrations, volatility distributions, and charge fractions with CS, reagent ions, and their corresponding neutral molecules.

The diurnal variations of cluster ions, their corresponding neutral molecules, and the charge fractions were shown in Fig. 7. The diurnal variations of cluster ions and their corresponding neutral molecules in Figure 7a-b reflect the significant role of neutral molecule variations in determining the cluster ion concentration, and the diurnal variations of charging fraction Figure 7c reflect a charging competition between  $H_2SO_4$  and organic ions.

Transportation of organic ions is considered to be limited due to their short lifetime, and the analysis of seasonality would need a longer measurement period.

Through the comprehensive quantitative analysis of atmospheric cluster ions, reagent ions, and neutral molecules, we can roughly predict the composition and concentration variations of most atmospheric negative cluster ions from measured neutral molecules and CS, as indicated by Fig. 8. However, as the actual charging fractions for different species differ in a wide range as shown in Fig. 9, the prediction of actual concentrations of atmospheric ions would require more work.

**(5) Lines 330-331: I don't see how this implies that ion-induced nucleation is stronger under clean conditions. Please provide further analysis supporting this statement.**

**Response:** Thanks for the suggestion. Here we meant that the concentration of NPF-related ion clusters including  $(H_2SO_4)_0$ .  $_2HSO_4^-$ ,  $(H_2SO_4)_3NH_3HSO_4^-$ , and  $(H_2SO_4)_3C_2H_7NHSO_4^-$  all increased significantly during clean periods, indicating enhanced ion-induced nucleation. As its relationship with our main topic is weak, this sentence has been removed from the revised manuscript.

**(6) Lines 332-334: Is the nitrogen-containing organic ions referring to CHONNO3-? If so, I don't understand the point about a higher fraction in Beijing given that Figure 4a shows 27% for Beijing haze while the Hyytiälä pie chart shoes 31% suggesting a larger fraction in Hyytiälä. Moreover, from the pie charts in Fig. 4b, there also appears to be more CHON in Hyytiälä than in the Beijing haze.**

**Response:** We apologize for any confusion. 'nitrogen-containing organic ions' refers to the sum of CHON-NO3- and CHON-HSO4-. In the revised manuscript, we use CHON-related ions instead of the original expression. We meant that CHON-related ion fractions **among all organic ions** are more abundant in Beijing than in Hyytiälä, **not among all ions**. To avoid confusion, we revised the pie charts to show the factions of CHO-related ions and CHON-related ions in the total organic ions, as shown in Fig. 4 in the revised manuscript. It shows that the CHON-related ions account for over 70% of total organic ions in urban Beijing, and only 33% in Hyytiälä on average. The figure and manuscript have been revised as below,

Figure 4. The negative cluster ion compositions in Beijing and Hyytiälä.

[Line 335-341] "The negative organic ions are mainly CHO-related or CHON-related organic ions in the form of the adduct with NO3- or HSO4- (Fig. 4). Only minor fractions (~4%) are in the deprotonated form of CHO- or CHON-. These indicate that the ionization schematic of organic ions is mainly through ion-molecular reaction with NO3- or HSO4- in Beijing. As few neutral sulfur-containing organics were observed, it is unlikely that the identified CHON-HSO4- is a cluster of sulfurcontaining organics and NO3-, but rather a cluster of nitrogen-containing organics and HSO4-. Among all negative organic ions, CHON organic ions adducted with NO3- (CHON·NO3-) are the most abundant and account for 56% and 69% during clean and haze periods, respectively."

(7) Line 340: In (Bianchi et al., 2017) which are measurements also from Hyytiälä, HOMs were observed as adducts with HSO4- as well. This should be discussed. This paper is referenced in line 343, however it should be moved earlier and in the context of relatively little adduct formation with HSO4- observed in the dataset used for the comparison here. Furthermore, in the Bianchi et al measurements it appears that ~25-50% of the signal for a given HOM ion was from the HOM HSO4- adduct during the daytime – I'm not sure I would consider this as being particularly low.

**Response:** Thanks for the suggestion. We agree that during the daytime on the clear days of Hyytiälä, organic ion clusters charged by  $HSO_4^-$  are non-negligible. But the ratio between organic ions attached to  $HSO_4^-$  and  $NO_3^-$  would still be lower than that in urban Beijing. This is mainly due to the higher  $HSO_4^-/NO_3^-$  ratio in Beijing (~0.36) compared with Hyytiälä (~0.05), as illustrated in Fig. S13. We have modified the manuscript as below,

[Line 360-363] "Moreover, a larger fraction of the organic ions are in the form of adducts with  $HSO_4^-$  in Beijing than in Hyytiälä. This is related to the higher  $HSO_4^-/NO_3^-$  ratio in Beijing (~0.36) compared with Hyytiälä (~0.05). The fraction of  $HSO_4^-$  adducted ions would increase during clear days in Hyytiälä but its fraction is still lower than those in Beijing (Bianchi et al., 2017)."

(8) Line 341-342: I am not entirely convinced by this argument since in Hyytiälä the H2SO4HSO4- peak is relatively more intense compared to HSO4- than it is in Beijing. Additionally, in Hyytiälä the HNO3NO3- ion is more intense than the NO3- ion compared to Beijing. This seems to imply that further nuance needs to be considered such as relative binding strengths of HNO3 NO3- vs HOM NO3- adducts (and likewise for HSO4- adducts).

**Response:** Thanks for the suggestion. As shown in Table R1, we compared the ratio of  $HSO_4^{-}/NO_3^{-}$ ,  $HSO_4^{-}/(HNO_3)_{0-2}NO_3^{-}$ , and  $(H_2SO_4)_{0-2}HSO_4^{-}/(HNO_3)_{0-2}NO_3^{-}$  in Beijing and Hyytiälä. The ratio of  $HSO_4^{-}/NO_3^{-}$  and  $HSO_4^{-}/(HNO_3)_{0-2}NO_3^{-}$  is both significantly higher in Beijing than in Hyytiälä. In contrast, the ratio of  $(H_2SO_4)_{0-2}HSO_4^{-}/(HNO_3)_{0-2}NO_3^{-}$  in Beijing is only slightly higher than in Hyytiälä. This is due to the abundance of  $(H_2SO_4)_{1-2}HSO_4^{-}$  in Hyytiälä. Thus we think that the low fraction of organic ions containing  $HSO_4^{-}$  is attributed to the low ratio of  $HSO_4^{-}/NO_3^{-}$  or  $HSO_4^{-}/(HNO_3)_{0-2}NO_3^{-}$ .  $(H_2SO_4)_{1-2}HSO_4^{-}$  should contribute little to the charging of OOMs because their formation energies are much lower than those of organic ions charged by  $HSO_4^{-}$  (Herb et al., 2018). As the concentration of  $(HNO_3)_{0-2}NO_3^{-}$  is dominated by  $NO_3^{-}$  in Beijing, we mainly consider  $NO_3^{-}$  as the main reagent ions in the atmosphere in the following analysis.

Table R1. The ratio between (H2SO4)0-2HSO4- and (HNO3)0-2NO3- in Beijing and Hyytiälä

|          | HSO 4 - /NO 3 - | HSO4 - /(HNO3)0-2NO3 - | $(H_2SO_4)_{0-2}HSO_4^-/(HNO_3)_{0-2}NO_3^-$ |
|----------|-------------------------------------------------------------|----------------------------------------------|----------------------------------------------|
| Beijing  | 0.3554                                                      | 0.3037                                       | 0.4459                                       |
| Hyytiälä | 0.0473                                                      | 0.0240                                       | 0.3681                                       |

**(9) Figure 4: Was the Hyytiälä data corrected for the API transmission function? If not, these comparisons are not meaningful.**

**Response:** Thanks for the suggestion. In the revised manuscript, we quantified the APi-TOF data measured in Hyytiälä using the *in-situ* quantification method with NAIS and updated the spectra as in Fig.4. It is stated in the revised manuscript as below,

[Line 141-142] "For further analysis in this study, we quantified the negative cluster ions in both Beijing and Hyytiälä using the *in-situ* quantification method, as will be discussed in Section 2.2."

(10) Sect. 3.3: Similar to Sect. 3.2, I find the analysis in this section too superficial to provide meaningful insights that will improve our understanding of atmospheric chemistry/composition. However, compared to Sect. 3.2, I think this section would require less work to add new insights. With the results presented as averages, overly broad generalizations are made an the text mostly focuses on reporting results rather than discussing the implications of those results. For instance, reporting the average charge fractions as a range of 4 orders of magnitude (line 381) does not provide meaningful information given the significant diel pattern of H2SO4 and the resulting impacts of charge competition. It would be interesting to know what is influencing such a range. For instance, it would seem reasonable that perhaps HOMs present primarily at night (i.e. formed from NO3 radical chemistry) might have higher charged fraction because of less charge competition with H2SO4. Is there a diel dependence evident in this charged fraction? Similarly, one might expect a day vs night dependence to perhaps influence the NO3- vs HSO4- charging (line 385). Delving a little more into the details controlling these wide ranges would allow the reader to gain more generalized insight into the chemistry and would be more helpful in analysis of future datasets. Such an analysis need not focus on all the ions – targeting the a select few (i.e. most intense HOMs) would be appropriate. This section would also benefit from clearly articulating the major scientific conclusions the reader should learn. For instance, is there a something to be learned about specific conditions where API is capable of providing insight into diel variations of neutral HOMs versus when it cannot? Further specific minor comments about this section are listed below.

**Response:** Thanks for the suggestion. In the revised manuscript, we have added deeper discussions on ion formation, its influencing factors, and implications. The sections are thus reorganized. The influence of CS, reagent ions, and neutral molecules on the formation of negative clusters was analyzed in detail. The charge competition between  $H_2SO_4$  and neutral molecules was clearly shown, and the charge fractions of organic ions with different compositions increased significantly during nighttime. We further show that molecules with high carbon atom numbers and high average carbon oxidation state are more easily to be charged. The influence of reagent ions,  $NO_3^-$  and  $HSO_4^-$ , on the formation of organic ions is mainly through their concentrations, and  $HSO_4^-$  clusters with organic molecules more efficiently than  $NO_3^-$ , except for NPs. Thus it

is possible to predict the variation of negative cluster ions based on the measured neutral molecules and CS. Though the ratio between them is highly dependent on their species, the charge fractions observed in our study can be used as a reference for the other urban environments. In addition, it is possible to provide some insight into the differences in charging efficiency of organics measured by using  $NO_3^-$  and  $HSO_4^-$  as the reagent ions.

**(11) Lines 377-379: What is meant by "neutral molecules detected by the CI-API-LTOF containing the same formula of CHO and CHON are treated as their corresponding precursors"? Does that mean the CHO- and CHON- ions were used or is it the formulas after subtracting NO3- as the reagent ion? The latter seems more appropriate to me.**

**Response:** Sorry for the confusion. We meant the latter: the formula of neutral molecules detected by CI-APi-TOF was identified by subtracting one  $NO_3^-$ . We revised the sentence in the manuscript for clearance:

[Line 374] "The neutral molecules detected by CI-APi-TOF was identified by subtracting one NO3-."

**(12) Figure 5: Please explain the histograms in panel b in greater detail. Why are the histograms different intensities overall? Are these total formulas detected? The N atom distribution is not shown in Fig. S6.**

**Response:** Sorry for the typo. The 'Fig. S6' should be 'Fig. S9'. The histograms are the number of species containing a certain number of C and O atoms. The number of species differs between APi-TOF and CI-APi-TOF because the number of peaks identified in APi-TOF is less than that in CI-APi-TOF. To make it clear, we have modified the figures as below,